# Mahan excitons in room-temperature methylammonium lead bromide perovskites

Tania Palmieri [1,5], Edoardo Baldini [1,5✉], Alexander Steinhoff [2], Ana Akrap[3], Márton Kollár [4], Endre Horváth[4], László Forró[4], Frank Jahnke[2] & Majed Chergui [1✉]

In a seminal paper, Mahan predicted that excitonic bound states can still exist in a semiconductor at electron-hole densities above the insulator-to-metal Mott transition. However, no clear evidence for this exotic quasiparticle, dubbed Mahan exciton, exists to date at room temperature. In this work, we combine ultrafast broadband optical spectroscopy and advanced many-body calculations to reveal that organic-inorganic lead-bromide perovskites host Mahan excitons at room temperature. Persistence of the Wannier exciton peak and the enhancement of the above-bandgap absorption are observed at all achievable photoexcitation densities, well above the Mott density. This is supported by the solution of the semiconductor Bloch equations, which confirms that no sharp transition between the insulating and conductive phase occurs. Our results demonstrate the robustness of the bound states in a regime where exciton dissociation is otherwise expected, and offer promising perspectives in fundamental physics and in room-temperature applications involving high densities of charge carriers.

[1] Laboratory of Ultrafast Spectroscopy, Lausanne Centre for Ultrafast Science (LACUS), Institute of Chemistry and Chemical Engineering (ISIC), École Polytechnique Fédérale de Lausanne (EPFL), CH-1015 Lausanne, Switzerland. [2] Semiconductor Theory Group, Institute for Theoretical Physics, University of Bremen, Otto-Hahn-Alle 1, P.O. Box 330440, Bremen, Germany. [3] Group of Light Fermion Spectroscopy, Department of Physics, Université de Fribourg, 3 Chemin du Musée, 1700 Fribourg, Switzerland. [4] Laboratory of Physics of Condensed Matter, Institute of Physics (IPHYS), École Polytechnique Fédérale de Lausanne (EPFL), CH-1015 Lausanne, Switzerland. [5]These authors contributed equally: Tania Palmieri, Edoardo Baldini. ✉email: ebaldini@mit.edu; majed.chergui@epfl.ch

In the quest for highly efficient light-energy conversion schemes in semiconductors, the manipulation of excitons is playing an increasingly central role. In photovoltaics, the formation of bound electron–hole (e–h) pairs upon photon absorption has dramatic effects on the charge transport, hindering efficient e–h dissociation at heterojunctions. In light-emitting devices, strongly bound excitons are instead desirable to reach high e–h capture rates for radiative recombination[1,2]. With these perspectives in mind, it is key to reveal the strength of e–h correlations in different classes of semiconductors and clarify how excitons react to the stimuli present in optoelectronic devices.

One such stimulus is the free carrier density injected via chemical doping or photoexcitation. Tuning this parameter results in a plethora of single-particle and many-body phenomena that profoundly influence the excitonic states[3]. For example, exchange and correlation effects modify the underlying electronic structure through bandgap renormalization (BGR). The free carriers also perturb the energy- and momentum-dependence of the material's dielectric background, screening the exciton binding energy ($E_b$). Finally, the single-particle states become partially filled, which leads to the weakening of the exciton oscillator strength and to the Burstein–Moss shift of the absorption onset. As the carrier concentration increases further, exciton dissociation processes turn the exciton gas into an ionized e–h plasma (scenario (i) in Fig. 1a). This is the essence of the paradigmatic insulator-to-metal (Mott) transition of band semiconductors[4], which manifests itself with the broadening and complete suppression of the excitonic peak (Fig. 1b, blue curve) at the so-called Mott density ($n_M$)[5].

While this picture of the Mott transition is correct to a first approximation, in 1967 Mahan proposed that bound excitonic states can still survive at doping levels above $n_M$[6]. In his seminal paper, Mahan considered a degenerate Fermi sea involving one type of carriers, akin to a chemically-doped semiconductor (scenario (ii) in Fig. 1a). In this high-density phase, an excited e–h pair can still interact with the electron gas, giving rise to exciton-like bound states—the so-called Mahan excitons. These states cause a singularity of the absorption at the Fermi edge, enhancing the above-gap oscillator strength (Fig. 1b, red curve). The phenomenon is known as excitonic enhancement or Fermi edge singularity and it can also emerge in metals[7,8]. While the original prediction discussed the quasi-equilibrium of chemically-induced carriers[5], joint theoretical-experimental studies extended this framework to photoexcited systems by defining quasi-Fermi levels for the e–h plasma (scenario (iii) in Fig. 1a)[9–12]. As a result, Mahan excitons can still be defined when intraband cooling is complete and the carriers have established a quasi-equilibrium within the bands. This scenario holds in a window of excitation densities between $n_M$ and the density required for population inversion, over which the Mahan excitons are still influenced by the simultaneous action of single-particle and many-body effects. Eventually, the subtle interplay between these effects can also lead the correlated e–h system to exhibit a rich variety of collective phenomena, typically at low temperature. Notable examples include the emergence of correlated metallicity[13], the instability towards nonequilibrium excitonic insulator phases (involving the Bose condensation of photoexcited e–h Cooper pairs)[3,14], and the nucleation of e–h droplets[15,16].

Discovering unambiguous signatures of Mahan exciton physics would open intriguing avenues in many-body theory and applied research. However, despite extensive efforts, this problem is far from being settled. So far, the majority of studies have focused on chemically- or photodoped semiconductors using steady-state spectroscopies[9–12,17–22]. Under static conditions, conventional lineshape analysis fails to disentangle the concurrent non-linearities in the optical spectrum and advanced calculations are required to isolate the possible fingerprints of Mahan

excitons[21,23]. In chemically-doped materials, this analysis is further complicated by the presence of donors/acceptors, which introduce random impurity-like potentials and significantly broaden any feature stemming from Mahan exciton physics[10].

To overcome these limitations, a promising strategy is exploring a nonequilibrium time-dependent setting through pump–probe spectroscopy. This allows for discriminating different phenomena on the basis of their timescale and unveiling the possible persistence of excitonic correlations above $n_M$. Preliminary results on indirect bandgap materials (Si and Ge) at cryogenic temperatures reported the robustness of the Wannier excitons above $n_M$ by monitoring the 1s–2p intraexciton transition through ultrafast terahertz spectroscopy[24,25]. However, the low probe photon energy used could not assess the presence of any Mahan exciton-related physics, i.e. the enhancement of the absorption continuum at high energy. The case of direct-gap materials in time-resolved experiments has been even more elusive to date, with no clear evidence for the existence of Mahan exciton states above $n_M$[26–28]. Due to the reduced scattering channels for their excitonic states compared to indirect gap materials[29], direct bandgap semiconductors hosting strongly bound excitons are the ideal platform in the search for Mahan excitons.

Among all such direct-gap semiconductors, the organic-inorganic lead-halide perovskites have recently attracted huge interest due to their outstanding photovoltaic performances[30] and high photoluminescence quantum efficiencies[1,31–37]. Numerous studies focused on understanding the extent to which the exciton concept describes the photophysics of these systems[38–41]. In $CH_3NH_3PbI_3$, $E_b$ of the order of the thermal energy was reported at room temperature (RT), suggesting the coexistence of weakly bound excitons and free carriers[40,42,43]. In contrast, the smaller dielectric constant of $CH_3NH_3PbBr_3$ single crystals results in a well-resolved excitonic peaks at RT, with $E_b$ ranging from 60 to 72 meV[44–46].

Here, we use methylammonium lead tribromide ($CH_3NH_3PbBr_3$) single crystals at RT to reveal direct evidence for Mahan exciton physics through a combination of time-resolved spectroscopy and advanced many-body calculations. We accurately evaluate $n_M$ through theory, and track the optical response of the material at photoexcitation densities well above $n_M$. A broadband optical probe allows us to uncover the persistence of the Wannier exciton peak, as well as the birth of the excitonic enhancement. We rationalize these observations through a model based on the semiconductor Bloch equations (SBEs), which confirms the absence of a sharp insulator-to-metal transition in the system and witnesses a crossover between the two phases. These results highlight the crucial role of e–h correlations in a regime where complete exciton dissociation is otherwise expected, paving the way towards a deeper understanding of the many-body phenomena emerging in semiconductors upon intense illumination[47].

## Results

**Estimation of the nominal Mott density.** A common approach to evaluate $n_M$ that is used in semiconductor physics directly compares the exciton Bohr radius and the screening length induced by the free carriers. This approximate method yields for $CH_3NH_3PbI_3$ values of $n_M$ on the order of $10^{16}$–$10^{18}$ cm$^{-3}$[35,48,49]. A similar estimate can be given for our sample of $CH_3NH_3PbBr_3$ (Fig. 1c), setting for $n_M$ an upper limit of approximately $2 \times 10^{18}$ cm$^{-3}$ (see Supplementary Notes 1 and 2 for the estimation of $E_b$ and $n_M$, respectively). However, these values of $n_M$ do not account for many-body effects and are rather inaccurate. Therefore, only a sophisticated theory can provide a reliable description of the interaction between excitons and free carriers. To this aim,

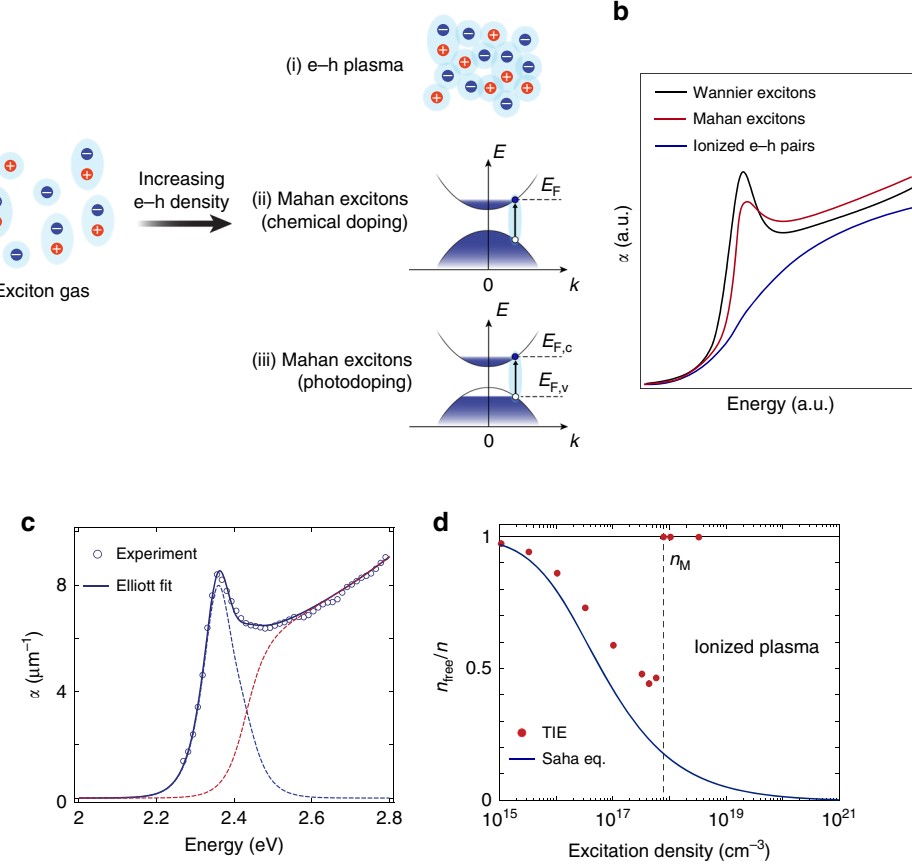

**Fig. 1 Absorption spectra and exciton ionization ratio. a** Evolution of the bound exciton gas in a bulk semiconductor with increasing carrier density. Two scenarios are possible: (i) bound excitons are ionized into an e–h plasma and the Mott transition to a metallic state takes place; (ii), (iii) e–h correlations still persist in the form of Mahan excitons, i.e. bound states in the Fermi sea in a (ii) chemically-doped and (iii) photodoped semiconductor. $E_F$ indicates the Fermi energy; $E_{F,c}$ and $E_{F,v}$ represent the quasi-Fermi energies of the conduction band and valence band, respectively. **b** Schematic representation of the optical absorption spectrum of a bulk semiconductor in the presence of Wannier excitons (black curve), and its modification at high carrier densities. The Mott transition manifests itself with the ionization of the Wannier exciton (blue curve), whereas the Mahan exciton scenario features the persistence of the Wannier peak and the enhancement of the absorption continuum (red curve). **c** Absorption spectrum of $CH_3NH_3PbBr_3$ single crystals as obtained from the ellipsometry data (dots), fitted with Elliott theory (solid line) and resulting in a binding energy $E_b = 71$ meV, linewidth $\Gamma = 34$ meV, and single-particle gap energy $E_g = 2.42$ eV. The blue and red dotted lines represent the distinct contributions of the Wannier exciton and the continuum, respectively. **d** Exciton ionization ratio as a function of the excitation density, where $n_{free}/n = 0$ corresponds to an exciton gas and $n_{free}/n = 1$ to a fully ionized plasma, as calculated from the theory of ionization equilibrium (TIE, red dots). The vertical line indicates the Mott critical density, found at $n_M \sim 8 \times 10^{17}$ cm$^{-3}$. The solid line represents the ionization ratio calculated with the Saha equation, and it is added for comparison.

here we use a many-body treatment of the exciton-plasma con-glomerate at a given temperature and excitation density using ionization equilibrium theory[50] (see Supplementary Note 3). It is based on the assumption that optically-excited e–h pairs in a semiconductor can form either a plasma of unbound (yet cor-related) carriers, or a gas of bound excitons, with all possible states of coexistence between the two phases depending on the experimental parameters. This approach yields the fraction of carriers forming the plasma ($n_{free}$) or the excitons ($n_X$). The fraction of ionized e–h pairs $\alpha_{eh} = n_{free}/n$, where $n = n_{free} + n_X$ is the total density of e–h pairs, is shown as red dots in Fig. 1d. At low carrier densities, $\alpha_{eh}$ closely follows the ionization ratio predicted by the Saha equation (solid line in Fig. 1d), which assumes the classical thermal equilibrium between free and bound pairs, and predicts the formation of excitonic species with increasing $n$ (see Supplementary Note 3). However, at about $5 \times 10^{17}$ cm$^{-3}$, the two curves abruptly deviate, and $\alpha_{eh}$ increases to 1. Many-particle renormalization and Coulomb screening (CS) lead to full exciton ionization at $n_M$ approximately $8 \times 10^{17}$ cm$^{-3}$ for $CH_3NH_3PbBr_3$ single crystals. $\alpha_{eh}$ is calculated using the experimental values for $E_b$, dielectric constant, and e–h effective

masses, i.e. the uncertainty in $n_M$ is related to that of these quantities, which remains below 10%.

**Ultrafast broadband optical spectroscopy.** Next, we map the ultrafast response of the material up to e–h densities above $n_M$. Specifically, we use intense above-bandgap excitation with an ultrashort laser pulse and monitor the pump-induced changes in the exciton optical response with a delayed continuum probe (details are given in the Methods section). Since the correct estimate of the photoexcited carrier density is central in this study, we precisely characterize all the uncertainties that may affect the experimental parameters (see Supplementary Note 4) and find that the error on the reported excitation densities is approximately 5%. A different choice of the excitation volume yields a variation in the density of less than a factor 2.

Figure 2a shows the color map of the transient reflectivity ($\Delta R/R$) as a function of probe photon energy and time delay between pump and probe for the excitation density of $5 \times 10^{18}$ cm$^{-3}$. We observe a derivative-like spectral response, which is positive above 2.40 eV and negative below. For time delays more than 1 ps, this signal agrees with previous reports at lower densities ($5 \times 10^{17}$ cm$^{-3}$),

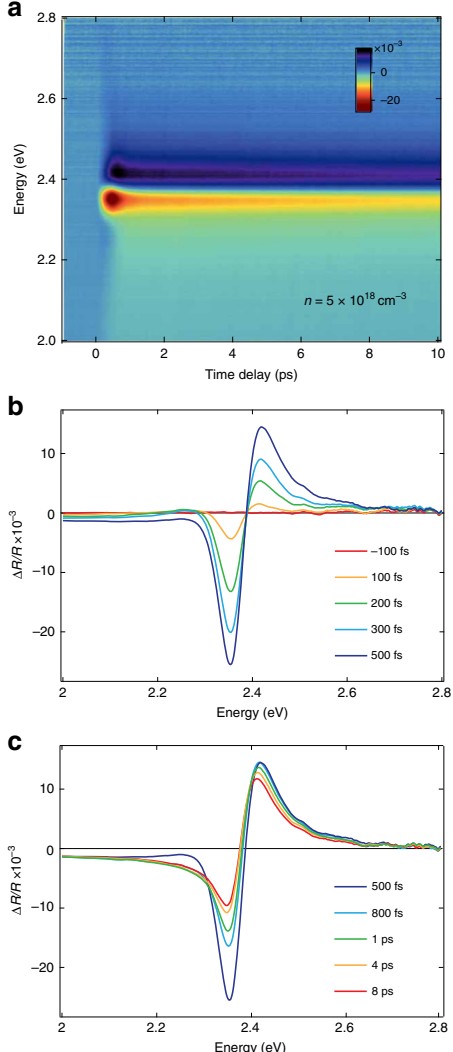

**Fig. 2 Ultrafast transient reflectivity measurements. a** Color-coded map of $\Delta R/R$ as a function of probe photon energy and time delay between pump and probe. The pump photon energy is 3.10 eV and the estimated carrier density is $n = 5 \times 10^{18}$ cm$^{-3}$. The time resolution is 50 fs. **b**, **c** $\Delta R/R$ transient spectra in the temporal windows **b** from −100 fs to 500 fs and **c** from 500 fs to 8 ps.

where it was assigned to the bleaching of the excitonic feature by band filling (BF)[51]. A closer inspection of our data reveals a complex evolution occurring in the first picoseconds. Specifically, the signal amplitude rises in the first 500 fs with an isosbestic point around 2.40 eV (Fig. 2b), while at later times it undergoes an asymmetric evolution around the inversion point, accompanied by a sizeable redshift of both the zero-crossing energy and the positive shoulder maximum (Fig. 2c). This trend is reproduced at different excitation densities above $n_M$ (5, 7.5 and $10 \times 10^{18}$ cm$^{-3}$, see Supplementary Fig. 4).

To retrieve the absorption spectra at the different excitation densities and time delays, we combine our steady-state and time-resolved optical data (see Supplementary Note 5). We model the reflectivity spectrum with a set of Lorentz oscillators, and describe its spectral changes through the variation of the model parameters. By iterating the fit for each time delay, we extract the time evolution of the oscillator strength ($\Delta OS$), peak energy ($\Delta E_X$), and linewidth ($\Delta \Gamma$) (Fig. 3a–c), as well as the time-dependent absorption coefficient $\alpha(\omega, t)$ (Fig. 3d, e). For the lowest density

($n = 5 \times 10^{18}$ cm$^{-3}$), we observe only a 10% decrease of the oscillator strength (Fig. 3a), accompanied by a slight redshift of $E_X$ (Fig. 3b) and a net increase of $\Gamma$ (Fig. 3c). All these effects occur in less than 500 fs. The same procedure is applied to the data measured at higher excitation densities (Supplementary Fig. 7), resulting in the absorption spectra in Supplementary Fig. 8. We observe that renormalization effects become more pronounced with increasing $n$; nevertheless, the Wannier exciton resonance remarkably persists, retaining a finite oscillator strength at all time delays. In addition, spectral weight is transferred from the exciton peak to the above-bandgap region, providing an enhancement of the absorption that has never been reported before. These results are summarized in Fig. 4a, which shows the RT absorption spectrum at three different excitation densities and at a representative time delay. The persistence of the excitonic feature in the absorption spectra up to $10^{19}$ cm$^{-3}$ is further confirmed by a separate lineshape analysis, presented in Supplementary Note 6. These observations support Mahan's prediction[6] and deviate from the conventional scenario of the Mott transition[4], in which an abrupt transformation from an insulating to a metallic phase occurs above $n_M$, along with a disappearance of the excitonic correlations in the absorption spectrum.

**Semiconductor Bloch equations.** In order to rationalize the fate of the excitons above $n_M$, we use the SBE[52–54] and calculate the absorption spectrum of $CH_3NH_3PbBr_3$ in the presence of photoexcited carriers. The SBEs describe the two-particle optical response based on the single-particle band structure, e–h occupancies, dipole matrix elements, and Coulomb interaction-induced bound states. Many-body effects that shape the optical response in the high-excitation regime, consistent with the theory of ionization equilibrium, can be systematically included. Since excitons are neutral particles, they do not cause significant BGR or CS. Thus, we consider only the influence of free carriers on the absorption spectrum. In addition, while spin–orbit splitting and Rashba effects are not explicitly included in the SBE, the effective masses utilized in our model originate from previous GW calculations that effectively account for the spin–orbit splitting[55]. More details are given in Supplementary Note 7.

The solution of the SBE under these assumptions yields the theoretical absorption spectra plotted in Fig. 4b for different photoexcited carrier densities (0.1 to $3.2 \times 10^{18}$ cm$^{-3}$). Here, the traces at the lowest densities are taken as a reference for the absorption coefficient at $n = 0$ cm$^{-3}$, since no sizeable changes are observed in the optical spectra up to $n$ approximately $3.2 \times 10^{17}$ cm$^{-3}$ (see Supplementary Note 7). The computed spectra also show the persistence of the excitonic feature above $n_M$ ($8 \times 10^{17}$ cm$^{-3}$), along with the enhancement of the above-bandgap absorption with increasing carrier density (black arrow), in excellent qualitative agreement with the experimental data. The theory predicts a blueshift of the absorption edge with increasing density that is also apparent in the experimental spectra. The size of the blueshift in the measured data (see Supplementary Fig. 8) originates from the different temporal evolution of the optical nonlinearities (BF, BGR, and CS) acting on the Wannier exciton peak. The mismatch between the results in Fig. 4a, b is governed by the assumptions underlying our theoretical model. In particular, the GW self-energy tends to overestimate renormalization effects, and our prediction of the critical density might be slightly smaller. In the SBE, the description of CS and BGR can be improved by introducing a vertex correction to the self-energy. However, such a correction is computationally challenging and results in a small variation of $n_M$, which should in any case not be larger than the experimental e–h densities.

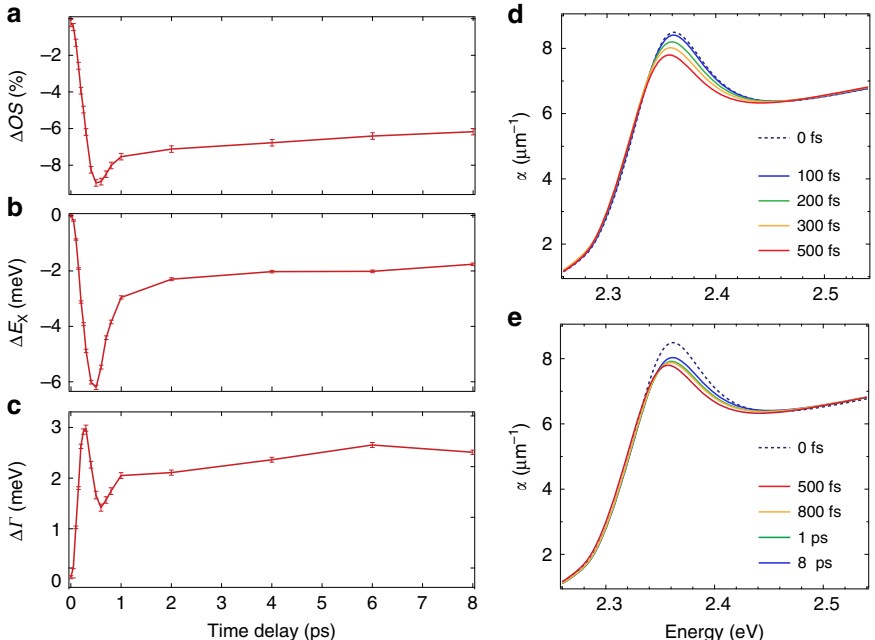

**Fig. 3 Reflectivity lineshape analysis. a–c** Temporal evolution of the oscillator strength ($\Delta OS$, **a**), peak position ($\Delta E_x$, **b**) and linewidth ($\Delta \Gamma$, **c**), obtained through the fit of $\Delta R/R(\omega, t)$ at the excitation density of $n = 5 \times 10^{18}$ cm$^{-3}$. **d, e** Evolution of the absorption spectra $\alpha(\omega, t)$ of CH$_3$NH$_3$PbBr$_3$ single crystals in the vicinity of the excitonic resonance as calculated with the Tauc-Lorentz fit of $\Delta R/R(\omega, t)$ in the temporal windows (**d**) from 0 fs to 500 fs and (**e**) from 500 fs to 8 ps.

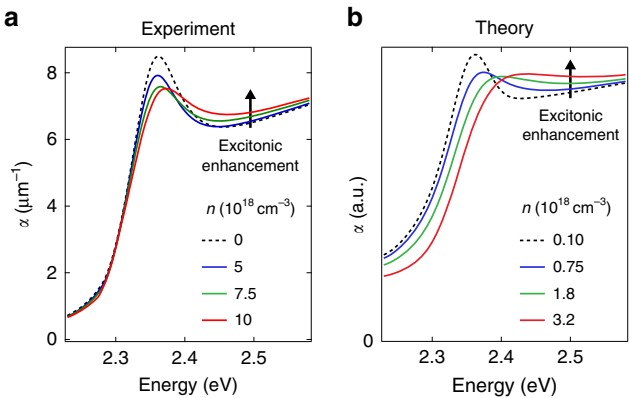

**Fig. 4 Comparison between experimental and theoretical absorption spectra. a** Experimental absorption spectrum of CH$_3$NH$_3$PbBr$_3$ single crystals for the excitation densities of 5, 7.5 and 10 $\times 10^{18}$ cm$^{-3}$ at 1 ps. Similar trends can be observed at any time delay between 500 fs and several ps. **b** Theoretical absorption spectra of CH$_3$NH$_3$PbBr$_3$ in the presence of increasing carrier densities, as calculated with the SBE. The black arrow indicates the excitonic enhancement of the above-gap absorption associated with the presence of e–h correlations in the highly photoexcited material. In both cases, the Wannier bound exciton peak persists above $n_M$.

Therefore, the qualitative agreement between theory and experiment is remarkable and it confirms the persistence of the bound states at the Mott transition contrary to their disappearance, which would result in optical properties resembling those of non-interacting e–h pairs (Fig. 1b, blue curve). The origin of the above-bandgap absorption increase can be explained by considering the distribution of e–h populations upon photoexcitation (scenario (iii) in Fig. 1a), where the excitonic enhancement arises from the strong e–h correlations in the Fermi sea above $n_M$. Therefore, our

calculations clarify that the Mahan excitons here detected do not stem from single electrons (holes) interacting with a degenerate Fermi sea of holes (electrons). Moreover, the robustness of the low-energy Wannier exciton provides additional evidence that the excitonic correlations are not completely screened in the highly-photoexcited state. These findings strongly indicate the existence of a crossover between the insulating and conductive phases in highly-excited CH$_3$NH$_3$PbBr$_3$ rather than a phase transition. In this scenario, e–h pairs still lead to exciton-like states, but in the form of Mahan excitons[5,12,21,25]. Finally, to investigate the role of the material parameters in this process, we solve the SBE for different values of dielectric constant (3.5, 7, and 10) and e–h effective masses (0.5, 0.75, and 1.5 times the effective masses indicated in Supplementary Note 2), which in turn yields different values of $E_b$ and $n_M$. The theoretical absorption spectra of CH$_3$NH$_3$PbBr$_3$ in the presence of increasing $n$ are shown in Supplementary Fig. 12. These results suggest that the presence of a large $E_b$ is key to the emergence of Mahan exciton physics, irrespective of whether this originates from a small dielectric constant or a large exciton effective mass (bottom graphs in Supplementary Fig. 12b, c). In contrast, for small $E_b$, the exciton peak promptly disappears as the e–h density overcomes $n_M$, indicating that under these conditions a Mott scenario still holds and the excitons become fully ionized.

We point out that there is no conflict between the exciton ionization ratio estimated in thermal equilibrium and the e–h correlations surviving above $n_M$ in the SBE solution. In fact, the theory of exciton ionization equilibrium is the most elaborate way to estimate the Mott density in a semiconductor. It assumes that the excited carrier density influences only the quasiparticle gap, but not the 1s-exciton energy. Under these conditions, $n_M$ is reached when the BGR compensates the exciton $E_b$. The SBE act as a complementary strategy that relies on the equilibrium state of the photoexcited carriers as an input. The SBEs can display the fate of possible excitonic correlations on top of the continuum by considering a density-dependent exciton energy in a particular parameter space of $E_b$ and $n$.

## Discussion

The present results show the peculiar excitonic character of the fundamental photoexcitations in $CH_3NH_3PbBr_3$ even beyond the densities at which, in the Mott scenario, the e–h plasma is expected to form. This observation is rather surprising and raises the question why similar fingerprints of Mahan excitons have not been reported before in other classes of direct-gap semiconductors. In the following, we ascribe our observations to the careful choice of experimental parameters in this study. First, the use of single crystals (instead of polycrystalline thin films) is beneficial to the visibility of the excitons, which are known to be extremely sensitive to grain boundaries and inhomogeneities[41]. Second, transient reflectivity represents the most accurate experimental methodology to map the existence and behavior of an exciton immediately after its creation. Alternative techniques like ultrafast photoluminescence would probe the exciton properties immediately after its annihilation, which is suitable neither to accurately track exciton ionization in semiconductors nor to identify the spectral signatures of Mahan excitons[56,57] (in particular the above-gap absorption enhancement). Third, the combination of large excitation densities above $n_M$ and RT: While lowering the temperature leads to a sharper quasi-Fermi surface, it also induces structural phase transitions[58], fine-structure splittings[59], and polaronic sidebands[60] that render the isolation of the spectroscopic fingerprint due to Mahan excitons very challenging. The final reason lies in the choice of the present material. Indeed, in the hybrid perovskite of the $CH_3NH_3PbI_3$ type (where $E_b$ is very small), a prompt disappearance of the bound exciton peak was reported using ultrafast THz spectroscopy and assigned to the dynamic screening of the exciton population[49], similarly to what observed in GaAs[61]. In $CH_3NH_3PbBr_3$, the large $E_b$ is key to hindering the full ionization of the exciton gas and therefore allows Mahan exciton-related spectral features to emerge at high carrier densities.

The present study supports a scenario in which a larger $E_b$ in a direct-bandgap semiconductor leads to a stronger persistence of the e–h correlations above $n_M$, irrespective of whether the enhanced $E_b$ stems from a small dielectric constant or from large effective masses. Our joint experimental-theoretical effort points toward rational-design strategies to search for signatures of Mahan exciton physics in other photoexcited direct-gap semiconductors with comparable $E_b$. Notable examples include wide-gap oxides such as ZnO ($E_b = 60$ meV) and nitrides such as GaN ($E_b = 30$ meV). These materials have been studied under steady-state conditions using chemical doping exceeding $n_M$. While features that could be attributed to Mahan excitons were highlighted[21,22], the spectra remained ambiguous because of inhomogeneities and the static nature of the measurement technique. In this respect, the application of pump–probe spectroscopies appears as a promising avenue to solve long-standing questions about the insulator-to-metal transition in band semiconductors. Preliminary results on highly-photoexcited ZnO seem to confirm this direction[62].

On the fundamental side, these findings deepen our knowledge of the exciton Mott transition and open intriguing possibilities towards the use of $CH_3NH_3PbBr_3$ as a suitable platform for Bose-Einstein condensation of Wannier exciton-polaritons in ad hoc designed microcavities[63]. On the applications side, the exceptional stability of the excitons highlights the potential of this hybrid perovskite in RT devices that demand high injected carrier densities, such as light-emitting devices and lasers[1,2].

## Methods

**Synthesis of $CH_3NH_3PbBr_3$ single crystals**. We synthesized high-purity crystals of $CH_3NH_3PbBr_3$ via solution growth as described by ref. [64]. Freshly cleaved surfaces of a few mm² were obtained by applying mechanical stress on the crystals with the tip of a blade. This results in the clean and shiny surface indicated by the black arrow in Supplementary Fig. 1a. The high crystallinity of the sample is demonstrated by the X-ray diffractogram in Supplementary Fig. 1b, which shows sharp lines for the oriented lattice planes and is free from the characteristic lines of degradation products.

**Spectroscopic ellipsometry measurements**. Steady-state spectroscopic ellipsometry measurements were performed to characterize the optical properties of the crystal at RT. The ellipsometric quantities $\Psi$ and $\Delta$ were measured for incident angles of 60°, 75°, and 78° in the energy range 1.00–5.50 eV, and used to obtain the complex dielectric function $\varepsilon = \varepsilon_1 + i\varepsilon_2$ (Supplementary Fig. 2a). To calculate the absorption spectrum (Fig. 1c) and the reflectivity (Supplementary Fig. 2b) we used the relations:

$$\alpha = \frac{2\kappa\omega}{c}, \qquad R = \frac{(1-n)^2 + \kappa^2}{(1+n)^2 + \kappa^2}, \qquad (1)$$

where $n$ and $\kappa$ are, respectively, the real and imaginary parts of the complex refractive index $\tilde{n} = n + i\kappa$:

$$n = \left(\frac{\varepsilon_1 + \sqrt{\varepsilon_1^2 + \varepsilon_2^2}}{2}\right)^{\frac{1}{2}}, \qquad \kappa = \left(\frac{-\varepsilon_1 + \sqrt{\varepsilon_1^2 + \varepsilon_2^2}}{2}\right)^{\frac{1}{2}}. \qquad (2)$$

**Transient reflectivity measurements**. Femtosecond broadband transient reflectivity ($\Delta R/R$) experiments were performed using the setup described in ref. [65]. A Ti: sapphire oscillator, pumped by a continuous-wave Nd:YVO₄ laser, delivers sub-50 fs pulses at 1.55 eV with a repetition rate of 80 MHz. The output of the oscillator seeds a cryo-cooled Ti:sapphire amplifier, which is pumped by a Q-switched Nd: YAG laser. The amplified laser system provides 45 fs pulses at 1.55 eV and a repetition rate of 6 kHz. One third of the output, representing the probe beam, is sent to a motorized delay line to set a controlled delay between pump and probe. The 1.55 eV beam is focused onto a 3 mm-thick CaF₂ cell using a combination of a lens with short focal distance and an iris to limit the numerical aperture of the incoming beam. The generated continuum covers the 2.00–2.80 eV spectral range. The probe is subsequently collimated and focused onto the sample through a pair of parabolic mirrors under an angle of 15°. The remaining two thirds of the amplifier output, representing the pump beam, is frequency-doubled to 3.10 eV with a β-barium borate (BBO) crystal and directed towards the sample under normal incidence. Along the pump path, a chopper with a 60 slot plate is inserted, operating at 1.5 kHz and phase-locked to the laser system. Both pump and probe beams are focused onto the sample on spots of dimensions 155 µm × 125 µm and 60 µm × 30 µm, respectively. The sample is mounted inside a closed-cycle cryostat. The reflected probe is dispersed by a fiber-coupled 0.3 m spectrograph and detected on a shot-to-shot basis with a complementary metal-oxide-semiconductor linear array. In a typical experiment, the acquisition of each data set is repeated multiple times to improve the statistics of the measurement. Hence, the experiment strongly relies on the repeatability of the scans, which requires stability of the sample under laser light illumination for several hours. Before the data analysis, the $\Delta R/R$ matrix has to be corrected for the group velocity dispersion of the probe. Since the probe beam is not dispersion-compensated after generation of the white light continuum, the probe pulses arrive at the sample stretched to a duration of few ps. This is beneficial for the experiment, because it significantly reduces the instantaneous probe light intensity on the sample. It is important to note that the probe beam dispersion is not a limiting factor for the time resolution of the setup, which is given on the detection side by the much smaller effective pulse duration per detector pixel. As such, the time resolution of the setup for all probe photon energies is 50 fs. Finally, we devoted particular attention in maintaining the sample under controlled environmental conditions to avoid its degradation or contamination. To this aim, all measurements were performed at RT, while keeping the crystal in the cryostat under a pressure less than $10^{-8}$ mbar[66,67]. The actual excitation densities impinging on the sample are calculated as $n = (1 - R)F/(h\nu\lambda_p)$ where $F$ is the pump fluence, $h\nu$ is the pump photon energy, $\lambda_p = 1/\alpha$ the penetration depth, and $R$ is the reflectivity of the sample. The final excitation density also takes into account the double reflection of the pump beam due to the cryostat window. All the parameters are evaluated at the pump photon energy of 3.10 eV.

## Data availability

The data that support the findings of this study are available from the corresponding authors upon reasonable request.

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

## Acknowledgements

The authors thank Prof. Giulia Grancini for useful discussions. T.P., E.B., A.A., and M.C. acknowledge financial support from the Swiss NSF via the NCCR:MUST, the contract

154056 (PNR 70, "Energy turnaround"), and the project PP00P2_170544. A.S. and F.J. acknowledge financial support from the Deutsche Forschungsgemeinschaft (RTG 2247 "Quantum Mechanical Materials Modelling") as well as resources for computational time at the HLRN (Hannover/Berlin). M.K., E.H., and L.F. were supported by the ERC Advanced Grant "Picoprop" (670918).

## Author contributions

T.P. and E.B. performed the transient reflectivity experiments, A.S. and F.J. performed the theoretical calculations. T.P. and A.A. performed the spectroscopic ellipsometry measurements, M.K., E.H., and L.F. prepared and characterized the sample. T.P. and E.B. analyzed the experimental data, T.P. and E.B. contributed to the data interpretation, T.P., E.B., and M.C. wrote the final manuscript. All authors participated in the final version of the article. T.P., E.B., and M.C. conceived the study.

## Competing interests

The authors declare no competing interests.
