## [Peer Review File · Nature Communications]

Reviewers' comments:

Reviewer #1 (Remarks to the Author):

In the present manuscript the authors performed the ultrafast spectroscopy to investigate the behavior of the exciton state in the methylammonium lead bromide perovskite under the high-density photoexcitation, with the analysis combined with the many-body theoretical calculations. They observed that the exciton line and enhanced above-gap excitation persist well above the Mott density, which they attribute to the formation of Mahan excitons.

The authors tackled the important problem which had long been discussed but remained unclear for many decades. The persistence of the exciton line at highest density observed here is indeed interesting, but I am not really convinced that it is attributed to the existence of Mahan excitons. I have three major questions/problems which should be clarified.

1) The result reported here is surprising only if the photoexcited density are estimated correctly. The authors calculate the density of the photoinjected carriers simply from the fluence of the pump light, accounting for the reflection loss with using the Fresnel factor. Can the authors exclude the saturation behavior of the photon absorption in this material under such a high-density excitation? Are the authors sure that the photoexcited carrier density is linear to the pump fluence in the whole excitation regime?

2) Throughout the manuscript, the authors assume that the existence of Mahan excitons will cause the persistence of the exciton bound state above the Mott density. However, to my knowledge, Mahan exciton is related to the Fermi edge singularity (FES) appearing in the optical spectrum, originating from the Coulomb interaction between the photoexcited electron (or the hole) and the degenerate hole (or electron) gas existing in the system, which is sometimes called "final state interaction". I am not aware of theory which discuss the robust exciton bound state in the storyline of Mahan exciton. It sounds misleading to me because FES does not necessarily require the well-defined bound state. Concerning this question,

2a) In the thermal equilibrium calculation shown as Fig. 1(d), the authors report the exciton ionization ratio of 1 above the $8 \times 10^{17} \text{ cm}^{-3}$. However, SBE calculation shown in Fig. 4(b) predict the persistence of the excitonic feature above that density, although it is broadened. It can be attributed to the electron-hole correlation surviving above the Mott density, but it is misleading to say that the exciton bound state persists because it conflicts with Fig. 1(d).

2b) It might be true that exciton bound state below the Mott density continuously connects to the FES (Mahan exciton effect) above the Mott density. In that case it is important to discuss how the band edge (E_g), chemical potential and exciton binding energy (E_b) behaves depending on the density. And it seems that authors can discuss those parameters. For example, the authors only show the peak position of the exciton line (E_x) in Fig. 3, but they should also show E_b and E_g .

3) In this paper, I cannot find a clear reason why this lead-bromide perovskite material is special to host Mahan excitons. The only reason the authors raise is the large exciton binding energy of 60-70 meV. However, it does not explain the reason why Mahan excitons are not observed in other materials.

3a) It is just a scaling problem, so Mahan exciton should be observed in other materials as predicted by SBE calculations which does not require any specific material characters, although it might be restricted to smaller density regions depending on the exciton binding energy of the materials.

3b) The authors discuss the reports on other doped wide-gap semiconductors ZnO and GaN whose exciton binding energies are also large. But the chemical doping has several problems as the authors discussed. Is there any report of the high-density photoexcitation experiments which support the Mahan exciton story?

In addition, I have smaller comments and questions,

4) In Fig. 4, the carrier densities of the experiment and calculation are considerably different. The

authors say "Furthermore, the quantitative agreement in the explored excitation densities between the two sets of data is limited." What is the reason the authors adopted the lower density results in the calculation? Is there any difficulty in the SBE calculation in the higher density region?

5) In the 10th line in the page 8, (Fig. 3a) and (Fig. 3b) appears in the wrong order in the text.

6) In the 14th line in the page 10, the authors say "our prediction of the critical density might be slightly smaller." Here, it is not clear what is "the critical density."

7) In the 15th line in the page 10, the authors say "experimental densities are also affected by uncertainty related to the laser fluence measurements." I do not think it will be so problematic. Instead, the estimation of the actually absorbed photon number (or here the penetration depth) is usually uncertain, with the error of factor 2-5.

To conclude, I would not recommend the manuscript to be published before the questions are correctly addressed.

Reviewer #2 (Remarks to the Author):

In this paper the authors claim observation of Mahan excitons in a $\text{CH}_3\text{NH}_3\text{PbBr}_3$ crystal. This claim is based on ultrafast pump-probe reflection measurements at room temperature, in combination with quantum many-body calculations of the optical spectra.

Mahan excitons are bound electron-hole states which exist in the presence of a degenerate electron or hole gas. A clear observation of Mahan excitons would be of high interest to a broad community.

The main question is now whether the presented data constitute a true observation of Mahan excitons. In my opinion, the presented data are far from convincing. The argumentation of the paper is that an excitonic feature is visible in the reflection spectrum at electron-hole densities above the Mott density, that is the density above which normal Wannier excitons do not exist as a result of Coulomb screening. I have a number of questions and comments about this argumentation.

1. In this paper the Mott density is determined to be $8 \times 10^{17} \text{ cm}^{-3}$. How accurate is this determination? The authors write that the error remains below 10%. But how can they claim such precision? It is well known, and acknowledged by the authors, that different types of calculations give different results, even differences of one or two orders of magnitude. Assumptions and simplifications are required, even if one uses semiconductor Bloch equations. Looking at the result of the semiconductor Bloch calculations (Fig. 4b) I see that there are still quite large differences with the experimental results (Fig. 4a).

2. How accurate is the experimental determination of the electron-hole density? This depends on the determination of the laser power and spot size and profile of the laser beam. How carefully have these parameters been determined? Also the absorption coefficient needs to be known with sufficient accuracy, as well as the amount of reflection at the sample surface, which could be affected by roughness or impurities. Furthermore, there may be fast decay of the charge carriers, which lowers the density.

3. The data which point at the existence of an exciton resonance above the Mott density could be explained by either an inaccuracy in the theoretical calculations, leading to a too low value for the Mott density, or an experimental inaccuracy, leading to a too high value for the electron-hole density. The authors should therefore find a way to experimentally distinguish Mahan excitons from the normal Wannier excitons.

4. Mahan excitons, as introduced by Mahan in Ref. 6 and as studied in Ref. 3, are the result of interaction between a hole and a degenerate electron Fermi gas, where the degenerate gas is the

result of doping. The present study describes experiments on a highly excited semiconductor, where both electron and hole gas are degenerate, as illustrated in Fig. 1. The physics is therefore different and seems to be more similar to the physics of BCS electron-hole pairs as studied in for example Keldysh and KopaeV, Sov. Phys. Solid State 6, 2219 (1965), Vasil'ev and Smetanin, PRB 74, 125206 (2006), Versteegh et al., 85, 195206 (2012) and Kim et al., Sci. Rep. 3, 3283 (2013).

5. According to Mahan (ref. 6), Mahan excitons depend on a sharp Fermi surface, limiting electron scattering, similar to the case of Cooper pairs in superconductivity. Therefore, Mahan excitons are more likely to occur at low temperatures. The authors have their sample already in a cryostat. Why do they not cool down and try to observe Mahan excitons at low temperatures?

6. Is there any special property of $\text{CH}_3\text{NH}_3\text{PbBr}_3$ which makes this material especially suitable for observing Mahan excitons?

7. The whole argument relies on the analysis of the transient reflectivity results, which are pretty complex. It would be good to have confirmation of the occurrence of Mahan excitons by using another detection method, for example photoluminescence or absorption.

In summary, I find the evidence presented in this paper by far not convincing enough for a claim of observation of Mahan excitons.

Reviewer #3 (Remarks to the Author):

The manuscript reports on the observation via transient absorption of Mahan excitons in the pump-induced change in reflectance of a MAPbBr_3 sample. I found the manuscript to be interesting but that the experimental observation of a Mahan exciton is not super convincing. But looks to be consistent with Mahan theory. I'm curious if the authors have thought about other effects that could exhibit similar changes to the reflectance. These are very high density excitations where non-linear effects become dominant.

It would be nice to see similar measurements on a system that is not expected to support Mahan excitons. Maybe they could repeat the measurements on GaAs where the exciton binding energy is a factor of 10 lower. What is special about the MAPbBr_3 sample? What would the temperature dependence look like? Additional measurements would help to support the authors' claims. Is there any difference between a Mahan exciton and an exciton that forms in a metal (see. Nature Physics volume 10, pages 505–509 (2014) for example). What is the significance of a Mahan exciton?

Reviewer #4 (Remarks to the Author):

The authors of the manuscript "Mahan excitons in room-temperature methylammonium lead bromide perovskites" combine accurate optical experiments based on ellipsometry and transient reflectivity measurements with insight from advanced many-body calculations involving the theory of ionization equilibrium and semiconductor Bloch equations to investigate possible Mahan excitons in highly excited organic-inorganic lead-bromide perovskites. The authors use cutting-edge techniques, both in experiment and theory, to obtain accurate data, allowing them to arrive at their conclusions. The material system studied here is of interest, e.g. for photovoltaics, and as stated by the authors, the precise absorption spectrum is important for such applications.

This clearly warrants investigation of the more fundamental question whether there is a Mahan exciton in this system and how it affects the optical absorption of this material. The present work is important in this context and cites relevant earlier papers, both experimental and computational work, appropriately. My feeling is that this manuscript touches on a timely topic and is indeed of fundamental as well as applied interest. The fundamental component is tied to the observation of

Mahan excitons specifically, but, more generally, also provides interesting insight into the interplay of phase-space filling, band-gap renormalization, and screening effects for this material on ultrafast time scales. Such insight into electron-electron and electron-phonon processes is important but only becomes accessible recently, with the emergence of ultrafast experimental techniques and computational approaches. The applied component of this work arises from the connection of optical properties of this important material and its applications. Hence, I believe that this manuscript is indeed of high interest for the broad readership of Nature Communications.

While I feel that this, generally speaking, justifies publication of this work in Nature Communications, I the authors should address the following questions prior to publication and prior to a final publication recommendation.

My main criticism of this work is that it is not obvious to me, how the present analysis, which is largely based on experimentally observed and theoretically predicted line shapes, goes beyond earlier studies (appropriately cited by the authors) regarding the unambiguous identification of the Mahan exciton that the authors emphasize. I assume that the key must lie in using pump-probe experimentation, but it is not clear to me how this adds crucial information that makes it more clear that the observed line shape is indeed a Mahan exciton. I feel that the authors should clarify how their methodology achieves a level of reliability when showing that the observed feature is a Mahan exciton, that previous studies didn't have. Unless I misunderstand, the authors also imply that in the case of a Mott transition, i.e. the absence of Mahan excitons, one would observe an independent-particle spectrum, as shown in Fig. 1b. Is this really the case, or would phase-space filling and band-gap renormalization still affect the line shape? Would, if such effects are present, that make the clear observation of Mahan excitons ambiguous?

Secondly, it would be helpful if the authors could clarify the relation of their observed line-shape features and Mahan excitons. In other words, is this really a Mahan exciton? The reason I am asking this question is that Mahan in his paper (cited by the authors) investigates doped systems with only one type of free carrier. Mahan then finds that in this situation bound states exist up to very high free-carrier concentrations and refers to this situation as a Mahan exciton. In the present work, however, the authors study a highly excited system where electrons and holes are present simultaneously. Photoexcited systems are not the same as degenerately doped systems and I am wondering whether the authors can comment on the transferability of Mahans results to this scenario?

In this context the authors should also clarify how they precisely describe screening effects. They state that "excitons can be viewed as neutral composite particles" and "do not induce significant renormalization or screening effects". How does the presence of excited electron and holes affect the formation of excitons in the theoretical framework used by the authors?

On a more minor note, it would be interesting if the authors could also comment on how well their model captures critical features of the band-structure of organic-inorganic lead-bromide, such as spin-orbit splitting and possible Rashba effects (if relevant for this structure). Does this matter for the conclusions drawn in this work?

Also, could the authors comment on their analysis of spectral features by fitting oscillators and comparing different probe delays? Is it crucial for the conclusions of this paper that the "character" of each oscillator stays the same as a function of probe delay, or is this not important (say, e.g., if different spectral features overlap and change differently in time)?

Finally, the color scale of Fig. 2a makes the plot itself almost obsolete, since practically no change is discernible as a function of time delay, especially compared to Figs. 2b and 2c.

There are two typos in the supplemental material: The first sentence of Note 5 refers to epsilon_2 twice (instead of epsilon_1 and epsilon_2). Also, the first sentence of the last paragraph starting on page 15 of the supplemental material is not clear to me.

Reviewer #1 (Remarks to the Author):

In the present manuscript the authors performed the ultrafast spectroscopy to investigate the behavior of the exciton state in the methylammonium lead bromide perovskite under the high-density photoexcitation, with the analysis combined with the many-body theoretical calculations. They observed that the exciton line and enhanced above-gap excitation persist well above the Mott density, which they attribute to the formation of Mahan excitons.

The authors tackled the important problem which had long been discussed but remained unclear for many decades. The persistence of the exciton line at highest density observed here is indeed interesting, but I am not really convinced that it is attributed to the existence of Mahan excitons. I have three major questions/problems which should be clarified.

We thank the reviewer for the careful reading of our work. We have provided a revised version of the manuscript and the Supplementary Information, which clarify the problems addressed by the reviewer and include his/her comments and suggestion.

1) The result reported here is surprising only if the photoexcited density are estimated correctly. The authors calculate the density of the photoexcited carriers simply from the fluence of the pump light, accounting for the reflection loss with using the Fresnel factor. Can the authors exclude the saturation behavior of the photon absorption in this material under such a high-density excitation? Are the authors sure that the photoexcited carrier density is linear to the pump fluence in the whole excitation regime?

As the reviewer points out, the precise estimation of the photoexcited carrier density is a crucial aspect of our work. We agree that it was not properly discussed in the previous version. We have now included a new Supplementary Note 4, where we estimate all sources of errors in the calculation of the experimental photoexcited carrier density and discuss the role of saturation effects. The error analysis leads to a maximum variation of the estimated fluence of a factor 2, which is intrinsic to the different possibilities in the choice of the excitation volume; such an uncertainty does not affect the comparison with the theoretical spectra nor the conclusion of our study. Concerning saturation effects, we observe a linear increase in the signal amplitude with increasing fluence (see Figures S4-S5), ruling out saturation in the observed response. While saturation effects are not significant in the excitation regime explored in our study, they are expected to appear close to the largest carrier densities explored (see *e.g.*, Saba et al., Nat. Comm. 5 (2014)).

Text added: Supplementary Note 4

2) Throughout the manuscript, the authors assume that the existence of Mahan excitons will cause the persistence of the exciton bound state above the Mott density. However, to my knowledge, Mahan exciton is related to the Fermi edge singularity (FES) appearing in the optical spectrum, originating from the Coulomb interaction between the photoexcited electron (or the hole) and the degenerate hole (or electron) gas existing in the system, which is sometimes called “final state interaction”. I am not aware of theory which discuss the robust exciton bound state in the storyline of Mahan exciton. It sounds misleading to me because FES does not necessarily require the well-defined bound state.

As the reviewer correctly pointed out, the emergence of a singularity (FES) in the absorption does not necessarily require the presence of a well-defined bound (Wannier) exciton state; however, the persistence of such a state above the critical Mott density provides additional evidence that the excitonic correlations are not completely screened in the highly photoexcited state. This Mahan-exciton scenario (where the exciton feature is still visible even as the binding energy of the Wannier exciton states becomes exponentially small) is thus opposed to the Mott scenario (in which full exciton ionization is expected). The persistence of the bound state and the excitonic enhancement are thus indicative that no Mott transition of the exciton occurs; instead, a Mahan-like exciton evolves in the system when the e-h density is increased (see Schleife et al., PRL 107 (2011) for a detailed description). We realized that this distinction was not clear in our discussion, and we modified the text accordingly.

Text added: pg. 12

Concerning this question,

2a) In the thermal equilibrium calculation shown as Fig. 1(d), the authors report the exciton ionization ration of 1 above the $8 \times 10^{17} \text{ cm}^{-3}$. However, SBE calculation shown in Fig. 4(b) predict the persistence of the excitonic feature above that density, although it is broadened. It can be attributed to the electron-hole correlation surviving above the Mott density, but it is misleading to say that the exciton bound state persists because it conflicts with Fig. 1(d).

We thank the reviewer for his/her comment, as it made us realize that this aspect has not emerged clearly in our previous version. The ionization equilibrium is the standard theory that is used to estimate the nominal Mott density in a semiconductor. In this framework, we assume that band-gap renormalization and screening of the exciton binding energy compensate each other, so that the absolute spectral position of the exciton remains the same in the presence of carriers. Therefore, the net effect of the carrier density is the reduction of exciton binding energy until it vanishes at the nominal Mott density, above which only unbound carriers are present.

A correct density-dependent renormalization of the exciton is instead included in the full solution of the SBE via the frequency-dependent correlation integrals (Eq. (19) and (20)), which consistently use the ionized fraction of carriers obtained by the theory of ionization equilibrium.

These calculations show that the exciton state persists and modifies the absorption line shape even if the exciton ionization theory predicts full ionization. As the two types of calculations complement each other, there is no conflict between the exciton ionization ratio estimated in thermal equilibrium and the statement that electron-hole correlations survive above the Mott density in the SBE calculations.

This aspect is rationalized by the calculation in static screening approximation by Schleife et al. PRL 107 (2011) which shows that, although the exciton binding energy becomes exponentially small, the bound state formally does not vanish. Reproducing these details by a very costly calculation is beyond the scope of our paper. Nevertheless, we included the above discussion in the new version of the manuscript.

Text added: pg. 12-13

2b) It might be true that exciton bound state below the Mott density continuously connects to the FES (Mahan exciton effect) above the Mott density. In that case it is important to discuss how the band edge (E_g), chemical potential and exciton binding energy (E_b) behaves depending on the density. And it seems that authors can discuss those parameters. For example, the authors only show the peak position of the exciton line (E_x) in Fig. 3, but they should also show E_b and E_g .

Following the reviewer's comment, we performed an additional lineshape analysis of the $\alpha(\omega, t)$ to retrieve the temporal evolution of E_b and E_g (see the new Supplementary Note 6 and Supplementary Figures 9-10). Although our modified Elliott model is not suitable for capturing the emergence of the FES nor the physics of semiconductors in the high-density regime, the reported trends for the Wannier exciton state (decrease in E_b and a stable/increasing oscillator strength, Supplementary Figures 9-10) deviate from the conventional scenario of the Mott transition, and further support Mahan's prediction of a high-density phase of strong e-h correlations.

Text added: Supplementary Note 6

3) In this paper, I cannot find a clear reason why this lead-bromide perovskite material is special to host Mahan excitons. The only reason the authors raise is the large exciton binding energy of 60-70 meV. However, it does not explain the reason why Mahan excitons are not observed in other materials.

The reviewer raises the crucial question why Mahan excitons have not been reported before in other classes of direct-gap semiconductors with strongly bound excitons, an aspect that was not clarified in the previous version of our manuscript. In the new version, we included a detailed paragraph in which we discuss how the choice of our experimental parameters is key to enable the observation of Mahan excitons. Briefly, these parameters are: i) The use of single crystals

over polycrystalline thin films, which is beneficial to the visibility of the exciton; ii) The use of a nonequilibrium transient (and not steady-state) technique such as transient reflectivity, which probes the exciton immediately after its creation (unlike photoluminescence, which is sensitive to exciton annihilation); iii) The creation of large excitation densities above the Mott density at room temperature, where the effects related to structural phase transitions, fine-structure splittings, and polaronic sidebands are not significant; this, in turn, facilitates the isolation of the spectroscopic fingerprint due to Mahan excitons. iv) The choice of the $\text{CH}_3\text{NH}_3\text{PbBr}_3$ over the $\text{CH}_3\text{NH}_3\text{PbI}_3$ compound, as a large E_b is key to allow Mahan exciton-related spectral features to emerge at high carrier densities. Further comments about this issue are given in the next answer to this referee.

Text added: pg. 13

3a) It is just a scaling problem, so Mahan exciton should be observed in other materials as predicted by SBE calculations which does not require any specific material characters, although it might be restricted to smaller density regions depending on the exciton binding energy of the materials.

We agree that as far as the SBE are concerned, the visibility of Mahan-exciton states would depend on a particular combination of parameters, and in particular on the exciton E_b . Following the referee's suggestion, we have solved the SBE for different values of dielectric constant and e-h effective masses (Supplementary Figure S12). These calculations confirm that the presence of a large E_b is key to the emergence of Mahan exciton physics, irrespective of whether this originates from a small dielectric constant or a large exciton effective mass. These results are of great interest for the theoretical description of many-body effects in excitonic material, and to our knowledge have never been addressed before. This aspect is discussed in the new version of the manuscript.

Changes made: pg. 12, Supplementary Figure S12

3b) The authors discuss the reports on other doped wide-gap semiconductors ZnO and GaN whose exciton binding energies are also large. But the chemical doping has several problems as the authors discussed. Is there any report of the high-density photoexcitation experiments which support the Mahan exciton story?

The concept of Mahan exciton is indeed transferable to photoexcited indirect and direct bandgap semiconductors. On the theory side, the description in terms of SBE of chemically-doped and photoexcited semiconductors is formally similar (Schleife et al., PRL 107 (2011)). On the experimental side, signatures of persisting excitonic correlations and/or enhancement of the continuum above n_M have been reported in photoexcited indirect-gap semiconductors (Grivickas

et al., PRL 91 (2003), Suzuki *et al.*, PRL 109 (2012), Asnin *et al.*, *Solid State Commun.* 47 (1983)) and direct-gap semiconductors (Richter *et al.*, arxiv.org/abs/1902.05832, Livescu *et al.*, *IEEE J. Quantum Electron.* 24 (1988), Olbright *et al.*, *Phys. Rev. Lett.* 66, 1358 (1991)). Although the fingerprints of Mahan excitons (reflected in the enhanced two-particle density of states) have emerged in the cited literature, a thorough study of this effect via systematic experimental measurements and theoretical calculations is still lacking. We have now discussed the issue of transferability in the revised version (modifying Fig. 1a accordingly and citing the relevant literature).

Text added: pg. 4

In addition, I have smaller comments and questions,

4) In Fig. 4, the carrier densities of the experiment and calculation are considerably different. The authors say “Furthermore, the quantitative agreement in the explored excitation densities between the two sets of data is limited.” What is the reason the authors adopted the lower density results in the calculation? Is there any difficulty in the SBE calculation in the higher density region?

Although there are no limitations to the inclusion of even higher densities, the salient features of our experiment are already successfully captured by the SBE in the density regime presented in the main text (from weak excitation to half an order of magnitude above n_M). Any deviation between the experimental and theory scales is due to approximations made in the theoretical description. Nevertheless, we believe that the agreement between theory and results goes beyond what is presented so far in any combined experimental-theoretical paper. We included these comments in the section of the main text devoted to the SBE calculations.

Text added: pg. 11

5) In the 10th line in the page 8, (Fig. 3a) and (Fig. 3b) appears in the wrong order in the text.

We thank the referee for highlighting this typo, we fixed it.

6) In the 14th line in the page 10, the authors say “our prediction of the critical density might be slightly smaller.” Here, it is not clear what is “the critical density.”

We corrected also these errors in the revised version of our manuscript.

7) In the 15th line in the page 10, the authors say “experimental densities are also affected by uncertainty related to the laser fluence measurements.” I do not think it will be so problematic.

Instead, the estimation of the actually absorbed photon number (or here the penetration depth) is usually uncertain, with the error of factor 2-5.

We agree with the reviewer regarding the estimation of the error in the experimental densities. The results of our error analysis, which were presented in a different answer to this reviewer's comments, show that the uncertainty in the impinging laser fluence is only 6%, while the maximum variation (a factor 2) is intrinsic to the different possibilities in the choice of the excitation volume. These aspects are discussed in the new Supplementary Note 4.

Text added: Supplementary Note 4

To conclude, I would not recommend the manuscript to be published before the questions are correctly addressed.

We thank the reviewer for her/his comments, which we believe we have addressed, leading to a significantly improved article.

Reviewer #2 (Remarks to the Author):

In this paper the authors claim observation of Mahan excitons in a $\text{CH}_3\text{NH}_3\text{PbBr}_3$ crystal. This claim is based on ultrafast pump-probe reflection measurements at room temperature, in combination with quantum many-body calculations of the optical spectra. Mahan excitons are bound electron-hole states which exist in the presence of a degenerate electron or hole gas. A clear observation of Mahan excitons would be of high interest to a broad community. The main question is now whether the presented data constitute a true observation of Mahan excitons. In my opinion, the presented data are far from convincing. The argumentation of the paper is that an excitonic feature is visible in the reflection spectrum at electron-hole densities above the Mott density, that is the density above which normal Wannier excitons do not exist as a result of Coulomb screening. I have a number of questions and comments about this argumentation.

We thank the reviewer for his/her valuable comments. We feel that the revised version of our manuscript now addresses all the points raised by the reviewer.

1) In this paper the Mott density is determined to be $8 \times 10^{17} \text{ cm}^{-3}$. How accurate is this determination? The authors write that the error remains below 10%. But how can they claim such precision? It is well known, and acknowledged by the authors, that different types of calculations give different results, even differences of one or two orders of magnitude. Assumptions and simplifications are required, even if one uses semiconductor Bloch equations. Looking at the result of the semiconductor Bloch calculations (Fig. 4b) I see that there are still quite large differences with the experimental results (Fig. 4a).

The referee is correct in saying that different types of Mott criteria give variations in the critical density of one or two orders of magnitude. For this reason, the calculation of the Mott density in our work is not based on such criteria, but on the theory of ionization equilibrium. This represents the most sophisticated approach, which builds on a many-body description of the exciton-plasma conglomerate; in this sense, it goes beyond the common Mott criteria that compare the exciton Bohr radius and the carrier screening length. Concerning the accuracy of the calculated value, some studies have claimed that vertex corrections are necessary to provide an accurate description of the absorption spectra in case of a Fermi edge singularity (Hawrylak, PRB 44 (1991) and references therein). Nevertheless, in the seminal paper, the concept of Mahan excitons are derived and explained based on the *only* use of ladder diagrams (Phys Rev. 153 (1967)), and the same level of theory is used by Schleife *et al.* to rationalize the experimental observation of Mahan excitons in chemically-doped ZnO (PRL 107 (2011)). In both cases, no vertex corrections are performed, and as far as we know, a calculation of exciton absorption spectra including diagrams beyond a dynamically-screened ladder approximation has never been done. In general, it is not possible to precisely predict the effect of vertex corrections for a specific situation; however, we expect that vertex corrections would be important at low

temperatures, but not at room temperature, where all singularities are smeared out. Including this into the theory would open a completely new topic in hybrid perovskite crystals, which goes beyond the scope of this paper.

Text added: Supplementary Note 3

2) How accurate is the experimental determination of the electron-hole density? This depends on the determination of the laser power and spot size and profile of the laser beam. How carefully have these parameters been determined? Also the absorption coefficient needs to be known with sufficient accuracy, as well as the amount of reflection at the sample surface, which could be affected by roughness or impurities. Furthermore, there may be fast decay of the charge carriers, which lowers the density.

We thank the reviewer for this comment. We included a new Supplementary Note 4, where we estimate all sources of errors in the calculation of the experimental photoexcited carrier density (laser beam parameters, optical coefficients of the samples, fast decay of charge carriers). We find that the absorbed excitation densities reported in our main text do not differ significantly from the actual value that is contributing to the observed response; moreover, a fast decay of the charge carriers can be ruled out as the intraband cooling time (500 fs) is much longer than our temporal resolution ($\ll 100$ fs).

Text added: Supplementary Note 4

3) The data which point at the existence of an exciton resonance above the Mott density could be explained by either an inaccuracy in the theoretical calculations, leading to a too low value for the Mott density, or an experimental inaccuracy, leading to a too high value for the electron-hole density. The authors should therefore find a way to experimentally distinguish Mahan excitons from the normal Wannier excitons.

This comment makes us realise that the previous version of our manuscript was confusing about the spectroscopic identification of the Mahan exciton fingerprint.

The primary signature of the Mahan exciton is given by the enhancement of the continuum (or Fermi edge singularity) above the exciton peak, which is due to strong e-h correlations in the Fermi sea; the persistence of the Wannier exciton provides additional evidence that the excitonic correlations are not completely screened in the highly photoexcited state. The persistence of the Wannier bound state and the continuum enhancement are thus indicative that no Mott transition (full exciton ionization) occurs above the critical Mott density; instead, a Mahan-like scenario is established, where the exciton feature is still visible even though the binding energy of the Wannier exciton becomes exponentially small (Schleife et al., PRL 107 (2011)). We clarified this distinction in the new version of the manuscript.

Text added: pg. 12

4) Mahan excitons, as introduced by Mahan in Ref. 6 and as studied in Ref. 3, are the result of interaction between a hole and a degenerate electron Fermi gas, where the degenerate gas is the result of doping. The present study describes experiments on a highly excited semiconductor, where both electron and hole gas are degenerate, as illustrated in Fig. 1. The physics is therefore different and seems to be more similar to the physics of BCS electron-hole pairs as studied in for example Keldysh and Kopayev, *Sov. Phys. Solid State* 6, 2219 (1965), Vasil'ev and Smetanin, *PRB* 74, 125206 (2006), Versteegh et al., 85, 195206 (2012) and Kim et al., *Sci. Rep.* 3, 3283 (2013).

The comment from the referee made us realise that the concept of transferability of Mahan exciton from chemically-doped to photo-doped systems did not clearly emerge in the previous version of our manuscript.

In fact, there is a vast literature that has extended the concept of Mahan exciton from doped to highly-photoexcited semiconductors. On the theory side, in the window of excitation density explored, the description in terms of semiconductor Bloch equations of chemically-doped and photo-doped semiconductors is formally similar (Schleife et al., *PRL* 107, 236405). On the experimental side, preliminary signatures of excitonic correlations and/or enhancement of the absorption continuum above the n_M have also been reported in photoexcited semiconductors and therein discussed theoretically (P. Grivickas et al., *PRL* 91 (2003), T. Suzuki et al., *PRL* 109 (2012), V. Asnin et al., *Solid State Commun.* 47 (1983), Richter et al, arxiv.org/abs/1902.05832, G. Livescu et al., *IEEE J. Quantum Electron.* 24 (1988), G.R. Olbright, et al, *Phys. Rev. Lett.* 66 (1991)). This concept is now clarified in our revised manuscript.

Text added: pg. 4

5) According to Mahan (ref. 6), Mahan excitons depend on a sharp Fermi surface, limiting electron scattering, similar to the case of Cooper pairs in superconductivity. Therefore, Mahan excitons are more likely to occur at low temperatures. The authors have their sample already in a cryostat. Why do they not cool down and try to observe Mahan excitons at low temperatures?

We agree with the referee that lowering the temperature would lead to a sharper quasi-Fermi surface, which would in principle favour an even more clear-cut observation of Mahan excitons. However, it would also induce structural phase transitions [A.D. Wright et al., *Nat. Commun.* 7 (2016)], fine-structure splittings [Baranowski, M. et al., *Nano Lett.* (2019)], and polaronic sideband effects [K. Miyata et al., *Sci. Adv.* 3, (2017)], making the isolation of Mahan excitons challenging and further complicating the line shape analysis. For these reasons, we chose to perform the experiments at room temperature (RT) while keeping the sample under vacuum in

the cryostat chamber to avoid contamination or degradation of our perovskite crystals. The RT study is also crucial in view of future optoelectronic devices operating in the high excitation density regime (LEDs and lasers).

This being said, studying the temperature dependence around the Mott transition represents a very interesting topic in many-body physics, and deserves to be explored in the future. We feel that our revised manuscript (which now integrates all the comments provided by this referee) represents a comprehensive study of the many-body effects occurring at RT in a photoexcitation regime, to our knowledge, hitherto not been investigated before.

Text added: pg. 14

6) Is there any special property of $\text{CH}_3\text{NH}_3\text{PbBr}_3$ which makes this material especially suitable for observing Mahan excitons?

We thank the reviewer for addressing this aspect. In the new version of our manuscript, we included a detailed paragraph in which we discuss how the choice of our experimental parameters is key to enable the observation of Mahan excitons. Briefly, these parameters are: i) The use of single crystals over polycrystalline thin films, which is beneficial to the visibility of the exciton; ii) The use of transient reflectivity as spectroscopic technique, which probes the exciton immediately after its creation (unlike photoluminescence, which is sensitive to exciton annihilation); iii) The creation of large excitation densities above the Mott density at RT, where the effects related to structural phase transitions, fine-structure splitting, and polaronic sidebands are not significant and facilitates the isolation of the spectroscopic fingerprint due to Mahan excitons. iv) Moreover, a crucial parameter is the choice of the $\text{CH}_3\text{NH}_3\text{PbBr}_3$ perovskite over the $\text{CH}_3\text{NH}_3\text{PbI}_3$ compound, as the large E_b is key to allow Mahan exciton-related spectral features to emerge at high carrier densities. We now further investigate this aspect by solving the semiconductor Bloch equations for different values of dielectric constant and e-h effective masses (Supplementary Figure S12), confirming that the presence of a large E_b is key to the emergence of Mahan exciton physics. These results are of great interest for the theoretical description of many-body effects in excitonic material, and to our knowledge have never been addressed before.

Changes made: pg. 12-13, Supplementary Figure S12

7) The whole argument relies on the analysis of the transient reflectivity results, which are pretty complex. It would be good to have confirmation of the occurrence of Mahan excitons by using another detection method, for example photoluminescence or absorption.

We find this comment by the reviewer particularly important, as in our original manuscript we did not discuss the reasons why we consider transient reflectivity as the most accurate detection method to unravel Mahan excitons.

In transient absorption spectroscopy, the quantity that is actually measured is the change in the transmission ($\Delta T/T$) of the broadband probe beam. The change into the transient absorption $\Delta\alpha$ (with the usual approximation $\Delta T/T = \exp(-\int_0^d \Delta\alpha dz) - 1$) is only possible when the sample reflectance is negligible in the spectral region of interest, which is not the case for our single crystal (see Figure S7). Therefore, performing a transient reflectivity ($\Delta R/R$) and a transient transmissivity ($\Delta T/T$) experiment would lead to the same type of data analysis, as both are complex optical quantities that depend on both the real and imaginary part of the dielectric function. However, transient transmission can be used only in the presence of thin films of hybrid perovskites, whose exciton visibility is significantly affected by the presence of domains, grain boundaries and inhomogeneities (Grancini et al., Nature photonics 9.10, 695 (2015)).

Concerning ultrafast photoluminescence (PL), this method does not accurately track exciton ionization in semiconductors, nor does it detect spectroscopic signatures of Mahan excitons (see H. Schweizer et al., PRL 51, 698 (1983)). Briefly, the Mott criterion indicates that the exciton binding energy vanishes at densities where the free carrier plasma is still non-degenerate. In this regime, the emission lineshape of the plasma becomes independent of the carrier density, hindering an accurate identification of the Mott density (if any). Furthermore, the theoretically-predicted fingerprint of Mahan excitons, *i.e.* an enhancement of the absorption continuum, cannot be observed by PL. The latter can in principle detect the persistence of the excitonic correlations in correspondence to the Wannier exciton resonance (similar to our reflectance experiment), but this does not imply the existence of Mahan excitons. However, as remarked by Toyozawa (Optical Processes in Solids, Chapter 10), the PL spectra would probe the Wannier exciton properties immediately before its annihilation, whereas reflectance/transmittance measurements map the existence and behaviour of an exciton immediately after its creation and thus constitute the ideal probe for unravelling the persistence of the Wannier exciton and the emergence of the Mahan enhanced absorption continuum before their decay and recombination.

Thanks to this referee, these points are now discussed in the new version of the manuscript.

Text added: pg. 14

Reviewer #3 (Remarks to the Author):

The manuscript reports on the observation via transient absorption of Mahan excitons in the pump-induced change in reflectance of a MAPbB₃ sample. I found the manuscript to be interesting but that the experimental observation of a Mahan exciton is not super convincing. But looks to be consistent with Mahan theory.

We thank the reviewer for his/her valuable comments. We have provided a revised version of the manuscript and the Supplementary Information.

1) I'm curious if the authors have thought about other effects that could exhibit similar changes to the reflectance. These are very high-density excitations where non-linear effects become dominate.

Yes, our interpretation was reached after careful consideration of conventional optical nonlinearities (band-filling, long-range Coulomb screening, and band-gap renormalization) that also contribute to the nonequilibrium optical response of a semiconductor. However, none of these effects can account for the enhancement of the absorption continuum above the bound exciton peak. We also considered the formation of additional bound states (biexcitons, trions, electron-hole droplets), but these effects are unlikely to occur at room temperature. Therefore, only from direct inspection of our spectra, we can conclude that formation of Mahan excitons is the most likely scenario that can account for the observed signal. We agree with the referee that these effects were not properly discussed in the previous version of our manuscript and we have modified the text accordingly.

Text added: Supplementary Note 6

2) It would be nice to see similar measurements on a system that is not expected to support Mahan excitons. Maybe they could repeat the measurements on GaAs where the exciton binding energy is a factor of 10 lower.

We thank the referee for the interesting comment. In fact, studies of plasma screening of excitonic correlations and related Mott density in low-binding energy direct gap semiconductors have been widely explored since the late 1970s: in this respect, extensive experiments and theoretical analysis have been performed on low temperature GaAs. Quasi-CW edge absorption measurements in the presence of above-gap photoexcitation (J. Shah et al., Phys. Rev. B 16, 1577) established the critical density at which exciton ionization occurs, and subsequent calculations (R. Zimmermann et al., Phys. Status Solidi B 90, 175) confirmed that the Mott criterion is indeed satisfied. In the case of GaAs, the critical Mott density is $5.4 \times 10^{14} \text{ cm}^{-3}$ at

low temperature; the transition is almost instantaneous upon photoexcitation, as observed in ultrafast pump-probe studies (one example is C. V. Shank et al., Phys. Rev. Lett. 42, 112). We address this comparison in the new version of our manuscript.

Text added: pg. 14

3) What is special about the MAPBr₃ sample?

We thank the reviewer for addressing this aspect, which was not clarified in the previous version of our manuscript. In the new version, we included a detailed paragraph in which we discuss how the choice of our experimental parameters is key to enable the observation of Mahan excitons. Among these, a crucial parameter is the choice of the CH₃NH₃PbBr₃ perovskite over the CH₃NH₃PbI₃ compound, as the large E_b is key to allow Mahan exciton-related spectral features to emerge at high carrier densities. We further investigate this aspect by solving the semiconductor Bloch equations for different values of dielectric constant and e-h effective masses (Supplementary Figure S12), confirming that the presence of a large E_b is key to the emergence of Mahan exciton physics. These results are of great interest for the theoretical description of many-body effects in excitonic material, and to our knowledge have never been addressed before.

Changes made: pg. 12-13, Supplementary Figure S12

4) What would the temperature dependence look like? Additional measurements would help to support the authors claims.

We agree with the referee that lowering the temperature would lead to a sharper quasi-Fermi surface, which would in principle favour an even more clear-cut observation of Mahan excitons. However, it would also induce structural phase transitions [A.D. Wright *et al.*, *Nat. Commun.* 7 (2016)], fine-structure splittings [Baranowski, M. *et al.*, *Nano Lett.* (2019).], and polaronic dressing effects [K. Miyata *et al.*, *Sci. Adv.* 3, (2017)], making the isolation of Mahan excitons challenging and further complicating the line shape analysis. For these reasons, we chose to perform the experiments at room temperature (RT) while keeping the sample under vacuum in the cryostat chamber to avoid contamination or degradation of our perovskite crystals. The RT study is also crucial in view of future optoelectronic devices operating in the high excitation density regime (LEDs and lasers).

This being said, studying the temperature dependence around the Mott transition represents a very interesting topic in many-body physics, and deserves to be explored in the future. We feel that our revised manuscript (which now integrates all the comments provided by this referee) represents a comprehensive study of the many-body effects occurring at RT in a photoexcitation regime, to our knowledge, hitherto not been investigated before.

Text added: pg. 14

5) Is there any difference between a Mahan exciton and an exciton that forms in a metal (see. Nature Physics volume 10, pages 505–509 (2014) for example). What is the significance of a Mahan exciton?

There is a subtle difference between a Mahan exciton forming in a highly-doped semiconductor and a certain kind of exciton in a metal, which is the different carrier density scale involved in the process. This said, in his seminal paper (Phys. Rev. Lett. 18, 448), Mahan discussed two cases for exciton formation in metals. One is similar to the semiconductor situation we have considered in the present work. The second is formed by a hole in the highest occupied band interacting with an electron excited to an even higher band. In this sense, there is no difference between these excitons since the many-body physics is described in terms of the same effects, such as bandgap renormalization, phase-space filling, and Coulomb screening.

The reference suggested by the referee is an interesting example where these effects can be generalized to the case of metal surfaces and, as such, we included it in our manuscript.

Reference added

Reviewer #4 (Remarks to the Author):

The authors of the manuscript "Mahan excitons in room-temperature methylammonium lead bromide perovskites" combine accurate optical experiments based on ellipsometry and transient reflectivity measurements with insight from advanced many-body calculations involving the theory of ionization equilibrium and semiconductor Bloch equations to investigate possible Mahan excitons in highly excited organic-inorganic lead-bromide perovskites. The authors use cutting-edge techniques, both in experiment and theory, to obtain accurate data, allowing them to arrive at their conclusions. The material system studied here is of interest, e.g. for photovoltaics, and as stated by the authors, the precise absorption spectrum is important for such applications.

This clearly warrants investigation of the more fundamental question whether there is a Mahan exciton in this system and how it affects the optical absorption of this material. The present work is important in this context and cites relevant earlier papers, both experimental and computational work, appropriately. My feeling is that this manuscript touches on a timely topic and is indeed of fundamental as well as applied interest. The fundamental component is tied to the observation of Mahan excitons specifically, but, more generally, also provides interesting insight into the interplay of phase-space filling, band-gap renormalization, and screening effects for this material on ultrafast time scales. Such insight into electron-electron and electron-phonon processes is important but only becomes accessible recently, with the emergence of ultrafast experimental techniques and computational approaches. The applied component of this work arises from the connection of optical properties of this important material and its applications. Hence, I believe that this manuscript is indeed of high interest for the broad readership of Nature Communications. While I feel that this, generally speaking, justifies publication of this work in Nature Communications, the authors should address the following questions prior to publication and prior to a final publication recommendation.

We thank the referee for her/his appreciation of our work. The main text and our Supplementary Information document have been modified accordingly to follow the referee's suggestion.

1) My main criticism of this work is that it is not obvious to me, how the present analysis, which is largely based on experimentally observed and theoretically predicted line shapes, goes beyond earlier studies (appropriately cited by the authors) regarding the unambiguous identification of the Mahan exciton that the authors emphasize. I assume that the key must lie in using pump-probe experimentation, but it is not clear to me how this adds crucial information that makes it more clear that the observed line shape is indeed a Mahan exciton. I feel that the authors should clarify how their methodology achieves a level of reliability when showing that the observed feature is a Mahan exciton, that previous studies didn't have.

This comment touches on the essence of our work and prompted us to emphasize why our experiment provides a better visibility of Mahan exciton physics than previous attempts.

In the following, we assume transferability of the Mahan exciton concept between chemically-doped and photo-doped systems, which will be discussed extensively below in a different answer (3) to this referee report.

First of all, we clarify that the primary spectroscopic fingerprint of the Mahan exciton is given by the enhancement of the continuum above the exciton peak, which is due to strong e-h correlations in the Fermi sea, while the persistence of the Wannier exciton provides additional evidence that the excitonic correlations are not completely screened in the highly-photoexcited state. Both features are indicative that no full exciton ionization occurs above the critical Mott density; instead, a Mahan-like scenario is established, where the exciton feature is still visible despite the binding energy of the Wannier exciton states becomes exponentially small (Schleife et al., PRL 107 (2011)). In the new version of the manuscript, we explain how the observation of such Mahan exciton fingerprints is enabled by the choice of our experimental parameters. Briefly, these include: i) the use of single crystals over polycrystalline thin films, which is beneficial to the visibility of the exciton; ii) the creation of large excitation densities above the Mott density at room temperature, where the effects related to structural phase transitions, fine-structure splittings, and polaronic sidebands are not significant and facilitates the isolation of the spectroscopic fingerprint due to Mahan excitons; iii) the choice of the $\text{CH}_3\text{NH}_3\text{PbBr}_3$ over the $\text{CH}_3\text{NH}_3\text{PbI}_3$ compound, as the large binding energy is key to allow Mahan exciton-related spectral features to emerge at high carrier densities (confirmed by the additional semiconductor Bloch equation calculations in Supplementary Figure S12).

As the reviewer correctly remarks, the use of transient reflectivity as spectroscopic technique is crucial, as one can probe the exciton immediately after its creation (unlike photoluminescence, which is sensitive to exciton annihilation). Moreover, transient reflectivity allows to disentangle the contributions of distinct nonlinear optical effects on the basis of their characteristic timescales, while steady-state spectroscopy methods would probe their superposition. In the absence of a direct spectroscopic signature, only the use of advanced theoretical calculations (A. Schleife et al., PRL 107, 236405) can help disentangling a possible Mahan exciton contribution from the other many-body phenomena.

Most previous works (cited in our manuscript) on Mahan excitons were performed at equilibrium on chemically-doped systems, or in photoexcited (but steady-state) in materials where a clear-cut fingerprint of persisting excitonic correlations above the nominal Mott density (e.g. incomplete disappearance of the bound exciton peak and emergence of the absorption continuum enhancement) was lacking. We think that the only experiment that provided a direct manifestation of incomplete disappearance of Wannier excitons is Suzuki *et al.*, PRL 109 (2012), which studied silicon *at low temperature*. Nevertheless, as this experiment was only sensitive to the low-energy spectrum of the sample, no evidence for the enhancement of the absorption continuum could be observed.

Changes made: pg. 12-13, Supplementary Figure S12

2) Unless I misunderstand, the authors also imply that in the case of a Mott transition, i.e. the absence of Mahan excitons, one would observe an independent-particle spectrum, as shown in Fig. 1b. Is this really the case, or would phase-space filling and band-gap renormalization still affect the line shape? Would, if such effects are present, that make the clear observation of Mahan excitons ambiguous?

Yes, single-particle (phase-space filling) and many-body effects (long-range Coulomb screening and band-gap renormalization) will affect the absorption lineshape in both the Mott and Mahan scenarios. In our work, the interpretation of the spectrum in terms of Mahan excitons was put forward after careful consideration of all the conventional high-density effects that contribute to the optical response of a semiconductor. However, none of these effects can account for the enhancement of the absorption continuum above the bound exciton peak. Therefore, only from direct inspection of our spectra, we conclude that the formation of Mahan excitons is the most likely scenario that can account for the observed phenomena.

This aspect is now addressed in the new version of the manuscript.

Text added: Supplementary Note 6

3) Secondly, it would be helpful if the authors could clarify the relation of their observed line-shape features and Mahan excitons. In other words, is this really a Mahan exciton? The reason I am asking this question is that Mahan in his paper (cited by the authors) investigates doped systems with only one type of free carrier. Mahan then finds that in this situation bound states exist up to very high free-carrier concentrations and refers to this situation as a Mahan exciton. In the present work, however, the authors study a highly excited system where electrons and holes are present simultaneously. Photoexcited systems are not the same as degenerately doped systems and I am wondering whether the authors can comment on the transferability of Mahan's results to this scenario?

Yes, the concept of Mahan exciton is transferable to highly-photoexcited semiconductors. On the theory side, the description in terms of semiconductor Bloch equations of chemically-doped and photo-doped semiconductors is formally similar (Schleife *et al.*, PRL 107 (2011)), and the same many-body effects (band-gap renormalization, screening, as well as phase-space filling) are at play. On the experimental side, signatures of excitonic correlations and/or enhancement of the absorption continuum above the n_M been reported in photoexcited indirect-gap semiconductors (Grivickas *et al.*, PRL 91 (2003), Suzuki *et al.*, PRL 109 (2012), Asnin *et al.*, *Solid State Commun.* 47 (1983)) and direct-gap semiconductors (Richter *et al.*, arxiv.org/abs/1902.05832, Livescu *et al.*, *IEEE J. Quantum Electron.* 24 (1988), Olbright *et al.*, *Phys. Rev. Lett.* 66, 1358

(1991)). Although the fingerprints of Mahan excitons (reflected in the enhanced two-particle density of states) have emerged in the cited literature, a thorough study of this effect via systematic experimental measurements and theoretical calculations was still lacking.

The concept of transferability of Mahan exciton from chemically-doped to photo-doped systems, which did not emerge clearly in the previous version of our manuscript, is now discussed in the revised version (modifying Fig. 1a accordingly and citing the relevant literature).

Text added: pg. 4

4) In this context the authors should also clarify how they precisely describe screening effects. They state that "excitons can be viewed as neutral composite particles" and "do not induce significant renormalization or screening effects". How does the presence of excited electron and holes affect the formation of excitons in the theoretical framework used by the authors?

In the Supplementary Notes 3 we show that the dynamical screening is treated in the random-phase approximation, considering contributions only from unbound carriers. In fact, due to the composite character of excitons, the screening contribution due to the bound carriers is not metal-like, but relies on the weak exciton dipole moment [Röpke et al., *phys. stat. sol. (b)* 92, 501 (1979)]. Similarly, the quasi-particle energy renormalizations due to excitons, as well as renormalizations of the excitons due to exciton-exciton interaction, are also very weak. In any case, due to the almost perfect neutrality of excitons, the exciton contribution to exciton screening and energy renormalization can be safely neglected with respect to the plasma contribution.

On the other hand, the role of the excited e-h in the exciton formation depends on the theoretical framework used. In the ionization equilibrium theory, the presence of electrons and holes leads to a band-gap renormalization and screening of the exciton binding energy. We assume that these effects compensate each other, so that the absolute spectral position of the exciton remains the same in the presence of carriers. Therefore, the net effect of the carrier density is the reduction of exciton binding energy until it vanishes at the nominal Mott density, above which only unbound carriers are present.

In the framework of semiconductor Bloch equations, the competition between quasi-particle renormalizations and screening is more subtle, since it is encoded in the frequency-dependent correlation integrals (eqs. (19) and (20) in the SI). These integrals consistently use the ionized fraction of carriers obtained by the theory of ionization equilibrium. However, unlike the latter, in the semiconductor Bloch equations there is no assumption of perfect compensation between the two effects, but all correlations are fully taken into account (on the level of screened ladder diagrams). In this sense, renormalizations of the excitons are described on a higher level as in the theory of ionization equilibrium. We realised that this aspect did not emerge clearly in the previous version of the manuscript and we modified it accordingly.

Text added: Supplementary Notes 3 and 7

5) On a more minor note, it would be interesting if the authors could also comment on how well their model captures critical features of the band-structure of organic-inorganic lead-bromide, such as spin-orbit splitting and possible Rashba effects (if relevant for this structure). Does this matter for the conclusions drawn in this work?

In our model, we could not include any spin-orbit splitting and Rashba effects explicitly. However, we used an effective mass approximation with masses taken from Sci. Rep. 6, 28618 (2016). In the cited work, spin-orbit coupling was effectively accounted for in the effective mass extracted from GW calculation. As such, we do not expect that these details have a significant impact on our results. Following the reviewer's comment, we added some sentences about this aspect in our revised manuscript.

Text added: pg. 11 and Supplementary Note 7

6) Also, could the authors comment on their analysis of spectral features by fitting oscillators and comparing different probe delays? Is it crucial for the conclusions of this paper that the "character" of each oscillator stays the same as a function of probe delay, or is this not important (say, e.g., if different spectral features overlap and change differently in time)?

No, it is not crucial that the character of each oscillator stays the same as long as the ensemble of the oscillators is capable of reproducing the evolution of the optical spectrum. More generally, any model for the dielectric function would be suitable as long as the ϵ_1 and ϵ_2 functions are Kramers-Kronig related and the model can describe both the steady-state dielectric function and the pump-induced changes in reflectivity. The advantage of our Tauc-Lorentz model is that the first oscillator can be used to track the changes in exciton peak position, linewidth and oscillator strength, which would not be possible if the peak was modelled with different functions. In our case, the final scope of the lineshape analysis is to access $\alpha(\omega, t)$ from $\Delta R/R(\omega, t)$ without the use of approximations and/or Kramers-Kronig transformations (which are inaccurate over a limited spectral range). Therefore, the main scope of the Tauc-Lorentz oscillators functions is to reproduce accurately the time-resolved spectral changes.

7a) Finally, the color scale of Fig. 2a makes the plot itself almost obsolete, since practically no change is discernible as a function of time delay, especially compared to Figs. 2b and 2c.

We changed the colour scales used in Fig. 2a accordingly.

7b) There are two typos in the supplemental material: The first sentence of Note 5 refers to

epsilon_2 twice (instead of epsilon_1 and epsilon_2). Also, the first sentence of the last paragraph starting on page 15 of the supplemental material is not clear to me.

We corrected these typos in the Supplementary Information.

Reviewers' comments:

Reviewer #1 (Remarks to the Author):

The authors have addressed most of the questions properly, though I still have one important and one minor comment.

The important one concerns the discussion on the thermal ionization equilibrium v.s. SBE to calculate the effect of excitonic bound state. I can understand the robust excitonic correlation survives above the Mott density, giving a Mahan exciton-like feature in the absorption spectra near the Fermi energy. And I have no problem with the statement that "the two type of calculations complement each other" as the authors say in the reply. However, the claim I can actually read in the revised manuscript, and in their detailed answer, is that the thermal ionization equilibrium is "imperfect" and SBE is more "correct" in terms of the treatment of the e-h correlation (For example, the authors put the words like "ionization equilibrium estimates the NOMINAL Mott density" and "CORRECT density-dependent renormalization"). If this is correct, the readers will feel that the experimental result shown here is not surprising, because it shows something beyond the "simple but inaccurate=thermal ionization equilibrium" model but consistent with "standard proper model=SBE". I do not really think this is correct, because I think the "exciton bound state" calculated in the ionization equilibrium and "Mahan type excitonic correlation effect" in the optical absorption spectra are essentially different objects. I think this point should be clarified before publication.

The minor comment is that I recommend the authors to mention about the electron-hole BCS scenario, and include some literature in this story line, as another referee mentioned. I agree on the transferability of Mahan exciton to photo-doped system, but on the other hand, the e-h BCS also has a long history of research.

After the authors addressed these points, I can recommend the publication in Nature Communications.

Reviewer #2 (Remarks to the Author):

The authors have improved their manuscript. They have clarified many aspects that were confusing and they have addressed most of my comments in an adequate way.

The main question to me was whether the presented data, in combination with the theoretical calculations, constitute convincing evidence for the existence of 'Mahan excitons' or excitonic enhancement at high electron-hole densities. My previous answer to this question was no. Now, after reading the revised manuscript with all the clarifications and additional data and arguments, I am willing to accept the presented data and calculations as evidence for Mahan excitons.

One worry that I had was the translation of the measured transient reflectivity into absorption spectra. It is a somewhat indirect way of measuring Mahan excitons. Can this be done with the required precision? And have the authors been able to measure the electron-hole density with sufficient accuracy? The data is not super striking and the analysis is complicated. Now, considering the answers from the authors, I feel that the authors have reached a reasonable level of accuracy to make conclusions on the absorption spectra.

Another worry that I had concerned the method to distinguish Mahan excitons from Wannier excitons. Now this has been clarified. The absorption in the continuum above the exciton peak distinguishes them.

Still, further improvements are necessary before the paper can be published. I am not satisfied with

the reply of the authors to my question nr. 4 and I think that the authors should revise the introduction of their paper and give some more clarification about the nature of the Mahan excitons they measured.

1. The term 'Mahan exciton' is here used to describe a bound state in a photoexcited semiconductor. This is not standard terminology in literature. Common terms to describe this effect are 'excitonic enhancement' and 'Fermi-edge singularity'. The term 'Mahan exciton', following the original work of Mahan (Refs. 6 and 7), refers to the interaction of an excited electron with a Fermi sea of holes or the interaction of a hole with a Fermi sea of electrons, as in Fig. 1a_{ii}. To use the same term for the photodoped case, where there is a Fermi sea of electron as well as a Fermi sea of holes is new (Fig. 1a_{iii}) and good arguments should be given for this.
2. In connection to the previous point, are the Mahan excitons detected in this work the result of single electrons interacting with a degenerate Fermi sea of holes, single holes interacting with a degenerate Fermi sea of electrons, or a degenerate Fermi sea of electrons interacting with a degenerate Fermi sea of holes?
3. The references to previous work are at many places inadequate: a) The introduction mentions: '... subsequent work extended this framework to photoexcited systems by defining quasi-Fermi levels for the e-h plasma ... [9-12]', failing to acknowledge the experimental data, besides the theory, that is also discussed in these four papers. b) 'However, despite extensive efforts [1, 13, 14], this problem is far from settled. So far, the majority of studies have focused on chemically- or photo-doped semiconductors using steady state spectroscopic studies.' Here, the authors should acknowledge that Ref. 13 is not about steady state spectroscopy, but about time-resolved PL and absorption measurements and Ref. 14 is also not about steady state spectroscopy, but about time-resolved reflectivity spectroscopy, a similar method as has been used in the present paper. c) 'However, the low probe photon energy used could not assess the presence of any Mahan exciton-related physics, i.e. the enhancement of the absorption continuum at high-energy. The case of direct-gap materials has been even more elusive to date, with no evidence for the existence of any bound states above nM.' This statement is a bit misleading. There has been evidence reported in literature for, what is here called, Mahan exciton-related physics in direct-gap materials, but I miss here references to these works. There is for example the work of Skolnick et al., Phys. Rev. Lett. 58, 2130 (1987), who demonstrated evidence for Fermi-edge singularity in InGaAs-InP quantum wells. Versteegh et al., Phys. Rev. B 85, 195206 (2012), presented evidence for, what they call, preformed electron-hole Cooper pairs, which are bound excitonic states that exist above nM, resulting from a degenerate Fermi sea of electrons interacting with a degenerate Fermi sea of holes (and therefore related to the here discussed Mahan excitons in photodoped semiconductor). Kim et al., Sci. Rep. 3, 3283 (2013) showed evidence for superfluorescence from Fermi-edge singularity in GaAs.

Reviewer #3 (Remarks to the Author):

The revised version of the manuscript is much improved from the original version and the authors have addressed all of the comments from this reviewer. I can recommend prompt publication. The new manuscript describes the Mahan physics and importance of the work much better.

One remaining question was in how the authors went from transient reflectance to the transient absorption. They used what looks to be a complicated fitting procedure. Why not simply do a Kramers Kronig transformation and the calculated change in absorption to the linearly determined absorption (fig. 1c). Probably some complicated reason why this would not work. But it seems that this would be

more satisfying than using the more complicated fitting procedure described here (but I have no reason to think this is not correct so I still recommend the manuscript be published).

Reviewer #4 (Remarks to the Author):

The authors address all comments of all reviewers in a satisfactory manner and I find the revised manuscript is significantly improved, so that it is suitable for publication. However, I suggest the authors address the following minor requests for clearer language:

1. On page 7, the authors state that their results are "paving the way towards a deeper understanding of the exciton Mott transition in semiconductors". In the light of their results, I wonder if it is still possible to speak of an "exciton Mott transition". It seems they find that there is no such transition?
2. In a similar way, in the first sentence of the discussion on page 14, the authors mention the expectation of e-h plasma formation. Does this expectation still hold? Doesn't this rely on the assumption of the existence of the exciton Mott transition?
3. In the last sentence of the first paragraph on page 13, the authors state that "for small binding energies, the exciton peak promptly disappears as the e-h density overcomes the critical Mott density." Does this mean there is a Mott transition, or does this simply mean that the Mahan exciton is hard to observe?
4. There is a typo on page 12 ("bangap") and on page 13 ("that that").

Reviewer #1 (Remarks to the Author):

The authors have addressed most of the questions properly, though I still have one important and one minor comment. The important one concerns the discussion on the thermal ionization equilibrium vs. SBE to calculate the effect of excitonic bound state. I can understand the robust excitonic correlation survives above the Mott density, giving a Mahan exciton-like feature in the absorption spectra near the Fermi energy. And I have no problem with the statement that “*the two type of calculations complement each other*” as the authors say in the reply. However, the claim I can actually read in the revised manuscript, and in their detailed answer, is that the thermal ionization equilibrium is “imperfect” and SBE is more “correct” in terms of the treatment of the e-h correlation (For example, the authors put the words like “*ionization equilibrium estimates the NOMINAL Mott density*” and “*CORRECT density-dependent renormalization*”). If this is correct, the readers will feel that the experimental result shown here is not surprising, because it shows something beyond the “simple but inaccurate = thermal ionization equilibrium” model but consistent with “standard proper model = SBE”. I do not really think this is correct, because I think the “exciton bound state” calculated in the ionization equilibrium and “Mahan type excitonic correlation effect” in the optical absorption spectra are essentially different objects. I think this point should be clarified before publication.

We thank the Referee for her/his comment, as it made us realize that our message about the two types of calculations did not emerge clearly in our manuscript. Historically, the Mott transition in semiconductors has been treated by introducing different criteria that attempt to describe the dissociation of the “bound exciton states” (i.e. the Wannier excitons) into an electron-hole plasma. As these criteria are very inaccurate, the ionization equilibrium theory has been formulated. The ionization equilibrium theory is not inaccurate, since it treats excitons and free carriers as separate species with a level of sophistication that goes beyond the approximate Mott criterion and the Saha equation. Therefore, this theory does capture the Mott density. However, around the Mott density value, the central assumption about the spectral separability of bound and unbound carriers fails. Therefore, one introduces the SBEs as a complementary strategy that relies on the equilibrium state of the photoexcited carriers as an input. The SBEs can display the fate of possible excitonic correlations on top of the continuum in a particular parameter space of binding energy and carrier density. What is remarkable in our results is that we find a parameter space (e.g., large binding energy and high carrier density) for the SBEs in which the technique can capture the Mahan excitonic correlations that are emerging in our experimental data. In this sense, the Referee is right

that one theory is not more correct than the other and that the exciton bound state and the Mahan exciton are essentially different objects. Thanks to the Referee's comment, we clarified this point in our revised version and modified the language used.

Text modified: page 14

The minor comment is that I recommend the authors to mention about the electron-hole BCS scenario, and include some literature in this story line, as another referee mentioned. I agree on the transferability of Mahan exciton to photo-doped system, but on the other hand, the e-h BCS also has a long history of research. After the authors addressed these points, I can recommend the publication in Nature Communications.

We added references to the electron-hole BCS scenario, following what this and Referee #2 suggest. We believe that this addition now makes our story line thorough and comprehensive.

Text added: page 4 and Supplementary Note 6

Reviewer #2 (Remarks to the Author):

The authors have improved their manuscript. They have clarified many aspects that were confusing and they have addressed most of my comments in an adequate way. The main question to me was whether the presented data, in combination with the theoretical calculations, constitute convincing evidence for the existence of ‘Mahan excitons’ or excitonic enhancement at high electron-hole densities. My previous answer to this question was no. Now, after reading the revised manuscript with all the clarifications and additional data and arguments, I am willing to accept the presented data and calculations as evidence for Mahan excitons. One worry that I had was the translation of the measured transient reflectivity into absorption spectra. It is a somewhat indirect way of measuring Mahan excitons. Can this be done with the required precision? And have the authors been able to measure the electron-hole density with sufficient accuracy? The data is not super striking and the analysis is complicated. Now, considering the answers from the authors, I feel that the authors have reached a reasonable level of accuracy to make conclusions on the absorption spectra. Another worry that I had concerned the method to distinguish Mahan excitons from Wannier excitons. Now this has been clarified. The absorption in the continuum above the exciton peak distinguishes them.

We thank the Reviewer for appreciating our revised work and for her/his detailed review of our manuscript.

Still, further improvements are necessary before the paper can be published. I am not satisfied with the reply of the authors to my question nr. 4 and I think that the authors should revise the introduction of their paper and give some more clarification about the nature of the Mahan excitons they measured.

We agree that some improvements were needed in the introduction to correctly cite the relevant literature and give more details about the Mahan exciton picture. The comments made by this Reviewer have helped us improve the quality of our introduction. Here below we address all the remaining questions.

1. The term ‘Mahan exciton’ is here used to describe a bound state in a photoexcited semiconductor. This is not standard terminology in literature. Common terms to describe this effect are ‘excitonic enhancement’ and ‘Fermi-edge singularity’. The term ‘Mahan exciton’, following the original work

of Mahan (Refs. 6 and 7), refers to the interaction of an excited electron with a Fermi sea of holes or the interaction of a hole with a Fermi sea of electrons, as in Fig. 1aii. To use the same term for the photodoped case, where there is a Fermi sea of electron as well as a Fermi sea of holes is new (Fig. 1aiii) and good arguments should be given for this.

We agree with the Referee that Mahan's original prediction was given for metals and degenerate (i.e. chemically-doped) semiconductors. However, the term "Mahan exciton" can be used for photo-doped semiconductors because they share the same description as chemically-doped semiconductors in terms of Bethe-Salpeter and semiconductor Bloch equations. The main difference lies in the fact that one scenario involves electron and hole occupancies as nonlinearity in the two-particle equation, whereas the other scenario relies on only one of them. In order for the same description to hold, the electrons and holes in the photo-doped semiconductor must have reached quasi-equilibrium, which implies that the Mahan exciton feature has to be well defined *after* the intraband cooling is complete (as in our data). Band filling can be stronger in the photoexcitation case, possibly leading to population inversion, and separate electron and hole chemical potentials have to be defined instead of one global chemical potential. However, there is a window of excitation densities between the Mott transition and population inversion where the situation is analogous to the simple doping case. There are the same many-body effects taking place, which are band-gap renormalization and screening due to intraband polarization. Hence, while the technical realization of chemical doping is different from photo-doping, the theoretical description is very similar as long as effects due to donor and acceptor levels can be neglected. Thanks to the Referee, we clarified this nomenclature issue with some sentences in the introduction of our revised manuscript, as well as with a paragraph in the Supplementary Information.

Text added: page 4 and Supplementary Note 6

2. In connection to the previous point, are the Mahan excitons detected in this work the result of single electrons interacting with a degenerate Fermi sea of holes, single holes interacting with a degenerate Fermi sea of electrons, or a degenerate Fermi sea of electrons interacting with a degenerate Fermi sea of holes?

The Mahan excitons detected in our work are the result of a degenerate Fermi sea of electrons interacting with a degenerate Fermi sea of holes, all created by photoexcitation. This aspect also

emerges from our semiconductor Bloch equations calculations, in which the enhancement of the above-gap absorption can be associated with the distribution of e-h populations created upon photoexcitation. In the revised manuscript, we highlighted this aspect in a clearer way in the section on the semiconductor Bloch equations.

Text added: page 13

3. The references to previous work are at many places inadequate: a) The introduction mentions: ‘... *subsequent work extended this framework to photoexcited systems by defining quasi-Fermi levels for the e-h plasma ... [9-12]*’, failing to acknowledge the experimental data, besides the theory, that is also discussed in these four papers.

It was not our purpose to neglect the experimental side of these works, but we wanted to keep the introduction brief and sharp before discussing different classes of steady-state and transient experiments. In the revised version of our manuscript, we follow the Referee’s suggestion by acknowledging the joint theoretical-experimental aspects of those references, and calling them again in the description of previous experimental efforts in detecting Mahan excitons in semiconductors.

Sentence and citations modified: pages 4-5

b) ‘*However, despite extensive efforts [3, 13, 14], this problem is far from settled. So far, the majority of studies have focused on chemically- or photo-doped semiconductors using steady state spectroscopic studies.*’ Here, the authors should acknowledge that Ref. 13 is not about steady state spectroscopy, but about time-resolved PL and absorption measurements and Ref. 14 is also not about steady state spectroscopy, but about time-resolved reflectivity spectroscopy, a similar method as has been used in the present paper.

We thank the Referee to point this out, since the way the references were used in those two sentences was indeed very misleading. Our purpose was to use the first sentence to introduce how the problem is far from being settled (without specifying the details) and then elaborate the discussion by discriminating between steady-state spectroscopic studies and time-resolved ones. In the revised version, we restructured the way the references are cited in order to avoid any confusion to the reader. We divided the references between the steady-state and the time-resolved experiments.

Furthermore, previous reference 13 was in a wrong position in the paper: it is a time-resolved PL that is appropriate to discuss how PL is sensitive to both exciton and electron-hole plasma formation. We moved it to the discussion part in the appropriate position. We also moved previous reference 14 to the part in which we describe previous pump-probe experiments that searched for signatures of persisting excitonic correlations above the Mott density.

Text changed: pages 4-5 - Reference moved to page 15

c) *‘However, the low probe photon energy used could not assess the presence of any Mahan exciton-related physics, i.e. the enhancement of the absorption continuum at high-energy. The case of direct-gap materials has been even more elusive to date, with no evidence for the existence of any bound states above n_M .’* This statement is a bit misleading. There has been evidence reported in literature for, what is here called, Mahan exciton-related physics in direct-gap materials, but I miss here references to these works. There is for example the work of Skolnick et al., Phys. Rev. Lett. 58, 2130 (1987), who demonstrated evidence for Fermi-edge singularity in InGaAs-InP quantum wells. Versteegh et al., Phys. Rev. B 85, 195206 (2012), presented evidence for, what they call, preformed electron-hole Cooper pairs, which are bound excitonic states that exist above n_M , resulting from a degenerate Fermi sea of electrons interacting with a degenerate Fermi sea of holes (and therefore related to the here discussed Mahan excitons in photodoped semiconductor). Kim et al., Sci. Rep. 3, 3283 (2013) showed evidence for superfluorescence from Fermi-edge singularity in GaAs.

The Referee is right in that some important references were missing in our previous manuscript. We added them and reorganized the way they are cited in the introduction. The initial presentation of the Mahan exciton problem is now followed by a list of experiments performed under steady-state conditions and finally by the time-resolved studies. We also mentioned the emergence of collective phases in order to account for the interesting comment provided by the Referee on electron-hole Cooper pairing. As correctly stated by the Referee, the Mahan exciton formation is related (as a possible precursor) but not identical to the preformation of electron-hole Cooper pairs that can eventually give rise to a nonequilibrium (Keldysh) excitonic insulator phase. Indeed, when exciton condensation occurs in nonequilibrium conditions, the bands of a highly-excited semiconductor undergo an electronic structure reconstruction and many-body gaps open at the quasi-Fermi surfaces of electrons and holes (analogous to the many-body gap that opens at the Fermi surface of a metal in the case of a superconducting instability). The size of these many-body gaps is strictly dependent on

the value of the exciton binding energy and their presence quenches any Fermi-edge singularity in the two-particle spectral function measured by optical spectroscopy. The same scenario is expected when the electron-hole Cooper pairs are preformed above the transition temperature for exciton condensation (and a BEC mechanism is thus at play instead of a BCS type of condensation). The gaps would still have a finite size (thus removing any Fermi-edge singularity), but the macroscopic phase coherence would be lost. This concept is widely explored in the physics of superconductors, where preformed pairing leads to a finite superconducting gap amplitude in the single-particle spectral function but long-range phase coherence is lost (an example for cuprates is given by T. Kondo et al., Nat. Phys. 7, 21-25). Similar results have been obtained for candidate equilibrium excitonic insulators close to the BEC regime (e.g., Ta₂NiSe₅ as reported by Y. Wakisaka J. Super. Nov. Magnet. 25, 1231-1234). In the paper by M. A. M. Versteegh et al. (Phys. Rev. B 85, 195206 (2012)), the authors correctly state that the pseudogaps in the single-particle density of states cannot appear in their theoretical treatment because the influence of fluctuations in the pairing fields on the self-energies of the electrons and holes has been neglected. A thorough description of these effects is offered by S.A. Moskalenko, D. W. Snoke, “*Bose-Einstein Condensation of Excitons and Biexcitons*”, which we now cite in our revised version.

Text added: page 4 and Supplementary Note 6

Reviewer #3 (Remarks to the Author):

The revised version of the manuscript is much improved from the original version and the authors have addressed all of the comments from this reviewer. I can recommend prompt publication. The new manuscript describes the Mahan physics and importance of the work much better.

We thank the Referee for endorsing prompt publication of our work.

One remaining question was in how the authors went from transient reflectance to the transient absorption. They used what looks to be a complicated fitting procedure. Why not simply do a Kramers-Kronig transformation and the calculated change in absorption to the linearly determined absorption (fig. 1c). Probably some complicated reason why this would not work. But It seems that this would be more satisfying then using the more complicated fitting procedure described here (but I have no reason to think this is not correct so I still recommend the manuscript be published).

Following the Reviewer's comment we included a discussion in Supplementary Note 5, which further motivates our choice of the fitting procedure rather than the Kramers-Kronig transformation (KKT). Briefly, in order to calculate the complex refractive index ($\tilde{n}(\omega) = n(\omega) + i\kappa(\omega)$) from the reflectivity ($R(\omega)$), the KKT require the knowledge of the phase shift angle of the sample ($\theta(\omega)$) at each frequency ω . This angle is expressed by an integral of the function $R(\omega)$ calculated over the $[0;\infty)$ frequency range. Since $R(\omega)$ is measured over a finite range, approximations are required at both extremes. Therefore, this method is suitable only for materials that show negligible $R(\omega)$ outside the measured range, which is not the case for $\text{CH}_3\text{NH}_3\text{PbBr}_3$ (see Supplementary Figure 2). For this reason, we implemented a model for the $R(\omega,t)$ and the $\alpha(\omega,t)$ that is based on the measured complex dielectric function rather than on the reflectivity spectrum. The photoinduced changes in $R(\omega)$ are modelled through *simultaneous* changes in $n(\omega)$ and $\kappa(\omega)$, which are dynamically Kramers-Kronig-constrained and whose steady-state functions are known a priori. This method is more robust than the iterative KKT of $R(\omega)$ in $\tilde{n}(\omega)$, because the calculation of $R(\omega)$ from $\tilde{n}(\omega)$ only requires the punctual values of $n(\omega)$ and $\kappa(\omega)$ at a given frequency (see Eqs (1) and (2) in the Methods section). As no integrals are calculated in this procedure, no approximations are used, providing a precise way of retrieving any optical quantity with the highest level of accuracy.

Text added: Supplementary Note 5

Reviewer #4 (Remarks to the Author):

The authors address all comments of all reviewers in a satisfactory manner and I find the revised manuscript is significantly improved, so that it is suitable for publication.

We thank the Referee for appreciating our revised manuscript.

However, I suggest the authors address the following minor requests for clearer language:

1. On page 7, the authors state that their results are "*paving the way towards a deeper understanding of the exciton Mott transition in semiconductors*". In the light of their results, I wonder if it is still possible to speak of an "exciton Mott transition". It seems they find that there is no such transition?

We agree that this sentence is misleading, as it might suggest that we actually observe an exciton Mott transition rather than the persistence of exciton-like states. We rephrased this part in the revised version.

Text modified: page 7

2. In a similar way, in the first sentence of the discussion on page 14, the authors mention the expectation of e-h plasma formation. Does this expectation still hold? Doesn't this rely on the assumption of the existence of the exciton Mott transition?

We thank the Referee for highlighting this aspect. In the discussion incipit, the purpose was to introduce our observation of exciton-like species and put it in contrast to the scenario usually expected above the Mott transition. We agree that the sentence was misleading, and we rephrased it following the Reviewer's suggestion.

Text modified: page 14

3. In the last sentence of the first paragraph on page 13, the authors state that "*for small binding energies, the exciton peak promptly disappears as the e-h density overcomes the critical Mott*

density." Does this mean there is a Mott transition, or does this simply mean that the Mahan exciton is hard to observe?

By saying that "*the exciton peak promptly disappears*", we imply that for small binding energies the Mott transition does take place, and no clear Mahan exciton features are detected. However, in the context of the sentence, we agree that this aspect has to be clearly specified. We rephrased this part following the Reviewer's suggestion.

Text modified: page 13

4. There is a typo on page 12 ("bangap") and on page 13 ("that that").

We corrected these typos in the revised version.

Text modified: pages 12, 13

REVIEWERS' COMMENTS:

Reviewer #1 (Remarks to the Author):

The authors have improved the manuscript properly according to the questions and comments raised by the referees, and now the manuscript is acceptable for publication.

One remark: I recommend the authors to cite the recent work by F. Sekiguchi et al., Phys. Rev. Lett. 118, 067401 (2017), where they revealed that electron-hole correlation survives above the Mott transition density in bulk GaAs, i.e., the excitonic correlation endows the high-density and low-temperature photoexcited carriers with nature like strongly correlated metal. Although in that paper the Mahan exciton scenario is not discussed, the physics behind seems to be intimately related.

Reviewer #2 (Remarks to the Author):

The authors sufficiently addressed all comments from the reviewers. Therefore I recommend this manuscript for publication in Nature Communications.

Reviewer #1 (Remarks to the Author):

The authors have improved the manuscript properly according to the questions and comments raised by the referees, and now the manuscript is acceptable for publication.

One remark: I recommend the authors to cite the recent work by F. Sekiguchi et al., Phys. Rev. Lett. 118, 067401 (2017), where they revealed that electron-hole correlation survives above the Mott transition density in bulk GaAs, i.e., the excitonic correlation endows the high-density and low-temperature photoexcited carriers with nature like strongly correlated metal. Although in that paper the Mahan exciton scenario is not discussed, the physics behind seems to be intimately related.

We thank the Referee for her/his comments, which greatly improved the quality of our work. We agree that the suggested reference is completing the literature review, and it is now cited in the introduction.

Reviewer #2 (Remarks to the Author):

The authors sufficiently addressed all comments from the reviewers. Therefore I recommend this manuscript for publication in Nature Communications.

We thank the Referee for her/his appreciation of our work and for his valuable contribution to the improvement of our manuscript.